



# Large-Eddy Simulation Sensitivities to Variations of Configuration and Forcing Parameters in Canonical Boundary-Layer Flows for Wind Energy Applications

Jeffrey D. Mirocha[1], Matthew J. Churchfield[2], Domingo Muñoz-Esparza[4*], Raj K. Rai[3], Yan Feng[5],
Branko Kosović[9], Sue Ellen Haupt[9], Barbara Brown[9], Brandon L. Ennis[6], Caroline Draxl[2], Javier Sanz Rodrigo[7], William J. Shaw[3], Larry K. Berg[3], Patrick J. Moriarty[2], Rodman R. Linn[4], Veerabhadra R. Kotamarthi[5], Ramesh Balakrishnan[5], Joel W. Cline[8], Michael C. Robinson[2,8], and Shreyas Ananthan[8#]

[1]Lawrence Livermore National Laboratory, Livermore, CA, 94550 – USA
[2]National Renewable Energy Laboratory, Golden, CO, 80401 – USA
[3]Pacific Northwest National Laboratory, Richland, WA, 99354 – USA
[4]Los Alamos National Laboratory, Los Alamos, NM, 87545 – USA
[5]Argonne National Laboratory, Lemont, IL, 60439 – USA
[6]Sandia National Laboratories, Albuquerque, NM, 87185 – USA
[7]Centro Nacional de Energías Renovables, Sarriguren, Navarra, 31621E – SP
[8]United States Department of Energy, Wind Energy Technology Office, Washington DC, 20585 – USA
[9]National Center for Atmospheric Research, Boulder, CO, 80305 – USA
[*]Current affiliation: National Center for Atmospheric Research, Boulder, CO, 80305 – USA
[#]Current affiliation: National Renewable Energy Laboratory, Golden, CO, 80401 – USA

*Correspondence to*: Jeffrey D. Mirocha (jmirocha@llnl.gov)

**Abstract.** The sensitivities of idealized Large-Eddy Simulations (LES) to variations of model configuration and forcing parameters on quantities of interest to wind power applications are examined. Simulated wind speed, turbulent fluxes, spectra and cospectra are assessed in relation to variations of two physical factors, geostrophic wind speed and surface roughness length, and several model configuration choices, including mesh size and grid aspect ratio, turbulence model, and
numerical discretization schemes, in three different code bases. Two case studies representing nearly steady neutral and convective atmospheric boundary layer (ABL) flow conditions over nearly flat and homogeneous terrain were used to force and assess idealized LES, using periodic lateral boundary conditions. Comparison with fast-response velocity measurements at five heights within the lowest 50 m indicates that most model configurations performed similarly overall, with differences between observed and predicted wind speed generally smaller than measurement variability. Simulations of convective
conditions produced turbulence quantities and spectra that matched the observations well, while those of neutral simulations produced good predictions of stress, but smaller than observed magnitudes of turbulence kinetic energy, likely due to tower wakes influencing the measurements. While sensitivities to model configuration choices and variability in forcing can be considerable, idealized LES are shown to reliably reproduce quantities of interest to wind energy applications within the lower ABL during quasi-ideal, nearly steady neutral and convective conditions over nearly flat and homogeneous terrain.



## 1 Introduction

Accurate characterization and prediction of the microscale wind flow environment plays an important role in many facets of wind power generation, including wind park siting, layout, operations, and the formulation of turbine design standards (e.g., Shaw et al., 2009), among others. While wind power generation has grown tremendously over the last few

decades, both turbine reliability and plant power generation frequently underperform projections based on existing turbine design standards and site assessments (e.g., Bailey, 2013). A key contributor to these underperformance issues is a disconnect between the data and models used in turbine and plant design and site assessment, and actual characteristics of the atmospheric boundary layer (ABL), the *in-situ* wind plant operating environment. Realistic ABL flows under routine atmospheric conditions often include much higher levels of atmospheric turbulence, shear, veer, and other important

transient phenomena than are typically captured in measurements or design tools.

Characterization of the wind plant operating environment has historically relied chiefly on observations, typically utilizing a small number of slow-response instruments, augmented occasionally by fast-response instruments capable of accurately characterizing turbulence (Magnusson and Smedman, 1994; Barthelmie et al., 2010). While remote sensing instruments (e.g., Högström et al., 1988; Barthelmie et al., 2003; Nygaard, 2011; Hirth et al., 2012; Rhodes and Lundquist,

2013; Smalikho et al., 2013; Iungo et al., 2013) provide one pathway to improve site characterization, the absence of fast-response turbulence information and limited sampling volumes provided by many systems, coupled with long deployments required to sample long-term variability, constrain the utility of observations for many applications.

Compounding the inadequacies of many observational datasets are the generally lower-fidelity numerical simulation approaches used in conjunction with observations to inform various stages of turbine and plant design and operation. While

higher-fidelity simulation techniques exist, their significant computational overhead has precluded widespread adoption due to limited computational infrastructure generally available to industry (Sanderse et al., 2011; Troldborg et al., 2011).

The increasing availability of high-performance computing infrastructure is enabling more widespread use of high-fidelity numerical techniques, such as turbulence-resolving large-eddy simulations (LESs), to significantly improve understanding of ABL and wind plant flows. While not yet considered as reliable as established observational and

computational approaches, high-fidelity numerical simulations can potentially provide superior site characterization and





design data to reduce costs, including 1) flow information over an entire wind farm across many levels within the turbine span, 2) simulation over a distribution of characteristic flow regimes in a short time period, and 3) estimates of flow parameters that are difficult or expensive to observe (e.g., turbulence).

While atmospheric LES is increasingly being utilized to simulate turbulent flows for wind energy applications (Sim et
al., 2009; Lu and Porté-Agel 2011; Bhaganagar and Debnath, 2014; Mehta et al., 2014; Mirocha et al., 2014a; Aitken et al., 2014), by focusing primarily on turbine wakes in quasi-ideal meteorological conditions, these studies have addressed only a limited range of atmospheric conditions and parameters of relevance to industry.  Development of atmospheric LES for general meteorological and surface conditions is ongoing, however this extension relies upon the development of novel forcing treatments both within the computational domain and at the lateral boundaries (e.g., Mirocha et al., 2014a; Muñoz-
Esparza et al., 2014, 2015), where assumptions of periodicity and standard approaches for specifying turbulent inflow conditions, such as recycling methods (e.g., Lund et al., 1998; Mayor et al., 2002), precursor simulations (e.g., Churchfield et al., 2012; Mirocha et al., 2014b), or synthetic turbulence generators (e.g., Veers, 1988; Jonkman and Buhl, 2005; Xie and Castro, 2008) are not applicable.

Irrespective of the complexity of the setup, high-fidelity atmospheric LES will require both thorough validation of
simulated quantities of interest, and formal assessment of uncertainties, prior to widespread adoption within the wind power industry. To satisfy these requirements, the Atmosphere to Electrons (A2e) initiative within the US Department of Energy's Wind Energy Technologies Office is supporting development and validation of next-generation computational approaches for wind energy applications. This is being undertaken via both assessment of existing simulation approaches, such as idealized LES, and development of new mesoscale-microscale coupling (MMC) methods, as required for extension to more
general environments and forcing conditions.

The present study, conducted under the auspices of the A2e MMC project, examines the efficacy of idealized atmospheric LES using periodic lateral boundary conditions (LBCs) to provide flow parameters of interest to wind energy applications. Examination of uncertainties is also undertaken, providing a required basis for assessment of both existing quasi-idealized simulation capabilities, as well as more sophisticated MMC techniques under development. Section 2

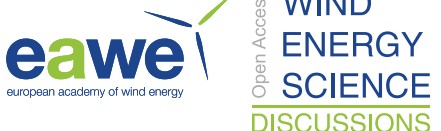



describes the case studies, code bases, boundary conditions, turbulence models, and variations employed to assess

uncertainty, Sect. 3 presents simulation results and uncertainty analysis, and Sect. 4 provides a summary and conclusions.

## 2 Methodology

Rather than focusing on turbine response and wake characteristics, as most studies of atmospheric LES targeting wind

energy applications have, we instead focuses on the accuracy of the resolved atmospheric flow field itself, including profiles

of wind speed and direction, turbulence kinetic energy, turbulent fluxes, and spectra and cospectra. Also included is

assessment of simulation uncertainties, undertaken by varying common numerical methods, turbulence models, and setup

approaches, using three simulation codes. The simulations are assessed against one another, theoretical expectations, and

observations taken from two case studies featuring quasi-ideal ABL flow during nearly steady neutral and convective

conditions over nearly flat and homogeneous terrain. The use of quasi-ideal conditions simplifies the attribution of

sensitivities to changes of various configuration and forcing parameters representing common simulation setups.

### 1.1 Case study selection

The location selected for the study is the Sandia Scaled Wind Farm Technology (SWiFT) test facility (Kelly and Ennis,

2016). Located on nearly flat and homogeneous terrain in the US Southern Great Plains, the SWiFT site possesses

characteristics that can be reasonably approximated in idealized, flat-terrain simulations using spatially uniform forcing and

surface characteristics, and periodic LBCs. A tower with fast-response instruments at five heights within the lowest 50 m

provided both mean flow and turbulence data. Nearby radar wind profiler data were used to assess the prevailing

meteorology, and to provide estimates of a common parameter used to force atmospheric LES, the geostrophic wind speed,

$U_g = \sqrt{u_g^2 + v_g^2}$ (with $u_g$ and $v_g$ the zonal and meridional components, respectively). Other parameters required to force the

simulations include the roughness length, $z_0$, which was estimated from the land cover, and fluxes of sensible heat, $H_S$,

measured at the tower base (latent heat flux was neglected).

To satisfy conditions under which idealized forcing is appropriate, data were examined for case studies encompassing

canonical neutral and convective regimes. Criteria for case selection included minimal mesoscale variability, nearly constant

values of wind speed and direction, and $H_S$, over time windows of a few hours, and minimal moisture-driven impacts.





While several periods approximating quasi-canonical convective ABL conditions appeared within the observational data, corresponding times for neutral conditions appeared relatively infrequently, and for shorter durations, during evening and morning transitions. Furthermore, sonic anemometers on the SWiFT tower are mounted on the booms pointing in the West-northwest direction while the dominant wind direction at the SWiFT tower is Southerly. As such, most of the candidate

canonical neutral conditions occurred during times at which the instruments were influenced somewhat by the tower wake.

From among approximately two years of available data, the periods selected to represent canonical nearly neutral conditions, given the multiple mitigating factors, occurred during the evening transition of August 17, 2012 following the waning of afternoon convection. The case study selected for convective conditions occurred during the early afternoon of July 4, 2012, during the apex of solar heating.

**1.2 Simulation code bases**

Three code bases representing standard approaches to ABL simulation are examined.

**1.2.1 WRF**

The Weather Research and Forecasting (WRF) model (Skamarock et al., 2008) is a community atmospheric simulation framework that supports applications ranging from global to micro scales, including LES, with several subfilter scale (SFS)

models available. WRF uses finite differencing to solve the compressible Euler equations, using a split time stepping algorithm within the Runge-Kutta time integration scheme, and a filter for acoustic modes. Advective discretization options include second- through fifth-order in the horizontal and second- or third-order in the vertical.

The WRF model uses a Cartesian mesh, with variables specified on an Arakawa "C" grid. Vertical spacing is specified using a terrain-following pressure-based eta coordinate.

At the model top, WRF imposes free-slip for $u$ and $v$, with vanishing $w$ and fluxes. For the studies herein, the surface shares the same Monin-Obukhov similarity approach as is applied within all three code bases, and described below.

**1.2.2 SOWFA**

The Simulator fOr Wind Farm Applications (SOWFA) (SOWFA, 2015) is a collection of flow solvers, turbulence SFS parameterizations, boundary conditions, and utilities for computing wind plant flows. SOWFA is built upon the Open-source



Field Operations And Manipulations (OpenFOAM) CFD Toolbox (OpenFOAM, 2015), a popular, open-source set of libraries for solving partial differential equations. OpenFOAM, hence SOWFA, uses an unstructured-mesh, finite-volume formulation for solving the governing equations. Several options exist for spatial discretization, with second-order central differencing typically used for the advective and diffusive terms. Time advancement is also second-order accurate with

Crank-Nicolson implicit discretization. SOWFA's flow solver is Boussinesq incompressible. All variables are located at cell centers, and to avoid velocity-pressure decoupling, a Rhie-Chow-like interpolation of velocity flux to cell faces is used. SOWFA includes Schumann's boundary condition for surface stress and additional boundary conditions for surface temperature flux or cooling rate.

### 1.2.3 HiGrad

The High Gradient applications (HiGrad) model (Sauer et al., 2016) discretizes the fully compressible, non-hydrostatic Euler equations using the finite-volume technique, on an Arakawa "A" grid. A variety of even- and odd-order advection schemes (first- to fifth-order accurate), as well as two LES SFS models, are available. A third-order explicit Runge-Kutta time-marching method is used in the present study.

### 1.3 Surface boundary conditions

For all simulations, the surface boundary condition is $w = 0$ with Monin-Obukhov logarithmic similarity theory (Monin and Obukhov, 1954) used to prescribe fluxes of momentum (with moisture ignored in these simulations) as

$$\tau_{i3}^s = -C_D U(z_1) u_i(z_1), \tag{1}$$

and heat,

$$H_S = -C_D [\theta_S - \theta(z_1)]. \tag{2}$$

Herein, $U$ is the scalar wind speed, $u_i$ are the resolved zonal ($i = 1$) and meridional ($i = 2$) velocity components, $z_1$ is the lowest computed height above the surface, and $\theta_S$ is the surface potential temperature, with $\theta = T(p_0/p)^{R/c_p}$, where $p_0 = 1 \times 10^5$ Pa is a reference value, $R$ is the gas constant for dry air, and $c_p$ is the specific heat of dry air at constant pressure.

$C_D = \kappa^2 \left[ \ln \left( \frac{z_1 + z_0}{z_0} - \psi \left( \frac{z}{L} \right) \right) \right]^{-2}$ in Eqs. (1) and (2) is the surface-atmosphere exchange coefficient, with $z_0$ the roughness

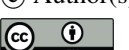



length, and $\psi\left(\frac{z}{L}\right)$ the stability function. During convective conditions, we follow Arya (2001) and Stull (1988) and use

$$\psi_M\left(\frac{z}{L}\right) = \ln\left[\left(\frac{1+\chi^2}{2}\right)\left(\frac{1+\chi}{2}\right)^2\right] - 2\tan^{-1}(\chi) + \frac{\pi}{2}, \text{ with } \chi = \left(1 - 15\frac{z}{L}\right)^{1/4}. \text{ Here } L = [-u_*^3\theta_{v0}]/[\kappa g H_S] \text{ is the Obukhov}$$

length, with $u_* = [(\tau_{13}^s)^2 + (\tau_{23}^s)^2]^{1/4}$, $\theta_{v0} = 300$ K a reference value of the virtual potential temperature, $\theta_v = \theta(1 + 0.61q_v)$, where $q_v$ is the water vapor mixing ratio, $\kappa = 0.4$ is the von Kármán constant, and $g$ is the gravitational

acceleration. For dry conditions, $q_v = 0$ and $\theta_v = \theta$. Due to the interdependence of $C_D$ and $L$ during non-neutral conditions, an iterative procedure is applied.

**1.4 Turbulence subfilter-scale (SFS) parameterizations**

Four SFS parameterizations were used in the sensitivity analysis (fuller descriptions are available in the references).

**1.4.1 Smagorinsky**

The Smagorinsky closure (SMAG) (Smagorinsky, 1963; Lilly, 1967) parameterizes the SFS stresses as $\tau_{ij} = -2K_M\tilde{S}_{ij}$, where $K_M = (C_S l)^2|\tilde{S}_{ij}|$ is the eddy viscosity coefficient for momentum, $C_S$ is the model constant, $l = (\Delta x_i)^{1/3}$ is a length scale, and $\tilde{S}_{ij} = \frac{1}{2}\left(\frac{\partial\tilde{u}_i}{\partial x_j} + \frac{\partial\tilde{u}_j}{\partial x_i}\right)$ is the resolved strain-rate tensor. Tildes denote resolved components of the flow, with $i = 1,2,3$ indicating the velocity components in the $x$- $(u)$, $y$- $(v)$, and $z$- $(w)$ directions, respectively.

Scalar fluxes are given by $S_j = -2K_q\frac{\partial\tilde{q}}{\partial x_j}$, with $K_q = Pr^{-1}K_M$ defining the eddy viscosity coefficient for scalar $q$, and

$Pr$ the Prandtl number. Default values utilized herein are $C_S = 0.18$ and $Pr^{-1} = 3$, with $l = (\Delta x\Delta y\Delta z)^{1/3}$. Modifications applied within WRF and SOWFA during stable conditions are ignored herein.

**1.4.2 Lilly**

The Lilly model (LILLY) (Lilly, 1967) is similar to the Smagorinsky closure, but uses $K_M = C_e l\sqrt{e}$, with $C_e = 0.1$ the model constant, and $e$ the SFS turbulence kinetic energy, obtained via integration of one additional prognostic equation (see

code description references for implementation details).



### 1.4.3 Nonlinear backscatter and anisotropy

The Nonlinear Backscatter and Anisotropy (NBA) model (Kosović, 1997) includes both a linear eddy viscosity

component, similar to SMAG and LILLY (but with different values for the constants), and a second term containing

nonlinear products of strain rate and rotation rate tensors. The NBA model can be formulated exclusively in terms of velocity

gradients (NBA-SR), or also using $e$ (NBA-TKE), with each dependent upon a single parameter, the backscatter coefficient

$C_b = 0.36$ (see above reference for details). As the NBA model specifies only the stresses, which directly determine

momentum, turbulent diffusion of scalars uses either the SMAG or LILLY closure, with $K_M$ diagnosed from the flow, and

used only to compute $K_q$.

### 1.5 Simulation setup

Simulations utilized domains of 2.4 km × 2.4 km × 2 km for the neutral case and 6 km × 6 km × 3 km for the convective

case, in the $x$-, $y$- and $z$-directions, respectively, with convective conditions requiring larger domains due to deeper ABLs

and convective rolls. Constant horizontal grid spacing is used for all simulations. The convective simulations used constant

vertical grid spacing throughout, while the neutral simulations used stretching (by 10 % per $\Delta z$) for $z > 500$ m.

While SOWFA and HiGrad use height as the vertical coordinate, WRF's use of a pressure-based coordinate precludes

exact specification of heights above the surface, therefore the heights of the pressure levels are initialized using the

hypsometric equation, $p(z) = p_S \exp\left(-gz/(R\overline{T})\right)$ (Holton, 1992), with $p_S$ the surface pressure, $R$ the dry air gas constant, $\overline{T}$

the standard atmosphere average temperature over a vertical layer of depth $\Delta z$, with $z$ the grid cell midpoint height value.

Subsequent changes to the thermodynamic state of the atmosphere during simulation can cause the heights corresponding to

the initial eta values to vary by a few percent over time.

Each simulation utilized damping in the upper portion of the model domain to prevent wave reflection at the model top.

WRF utilized Rayleigh damping, which nudges the horizontal wind components toward their geostrophic values, with a

coefficient value of 0.003 s$^{-1}$, and exponentially decreasing strength approaching the model top. HiGrad used a similar

Rayleigh damping function to that of WRF. SOWFA achieves damping in the upper region of the domain by smoothly

transitioning from using purely central differencing of the advective term to a mix of 90 % central and 10 % upwind above a





specified height. For all simulations, damping was applied within 400 and 600 m of the model top during the neutral and

convective simulations, respectively.

Simulations were initialized with thermodynamic variables approximating observations during the two case studies

described previously. Initial horizontal wind components were $u = u_g$, $v = v_g$ and $w = 0$, with potential temperature

profiles of $\theta(z) = \theta_B + a(z) + a'(z)$. Here, $\theta(z) = [p_0/p(z)]^{0.286}$, where $p_0$=1000 hPa is a reference pressure, $\theta_B$ is a

background constant value, $a(z)$ specifies an inversion, to prevent turbulence from reaching the model top, and $a'(z)$ are

small perturbations $\in [\pm 0.25\,K]$, drawn from a uniform distribution, and scaled as a decreasing cubic function of height

from the surface. These small perturbations are applied only to the initial condition to seed turbulence. The neutral

simulations used $\theta_B = 300$ K, with the termination of the perturbations and base of the inversion specified at 500 m, with an

inversion strength of 10 K km$^{-1}$. The convective simulations used $\theta_B = 309$ K, with perturbations up to 400 m, and an

inversion beginning at 600 m, with a strength of 4 K km$^{-1}$. SOWFA, which use temperature rather than $\theta$, specified initial

temperature to be consistent with $\theta$ as described above. WRF and HiGrad specified a hydrostatic base state pressure

distribution using the above described $\theta$ distribution and $p_S = P_0$. SOWFA, which uses an incompressible solver, and

therefore requires no background pressure (only gradients of pressure are resolved), added a background value of $p_0$ the

computed pressure field to be consistent with the other solvers.

For these idealized simulations, which were based upon case studies with no precipitation, and little cloudiness or

synoptic-scale weather variability, the simulations were initialized dry, and the only physical process parameterizations

utilized were SFS turbulence fluxes, with surface sensible heat and stresses as described in Sect. 1.3.

Due to the initial flow field being nonturbulent and not in balance with the applied geostrophic forcing, a spin-up period

was required for the flow statistics to approach nearly steady values. During neutral conditions, the spin-up period is longer,

due to the weak turbulence forcing, and existence of a strong inertial oscillation with a period of several hours (at the

specified latitude of 33.5 degrees). As differences in model formulation, mesh resolution, and turbulence SFS model all

influence the period of the inertial oscillation, via impacts on turbulent transport, the simulations were compared during the

two hours surrounding the point in time in which each simulation reached its first maximum in the planar, 10-minute average

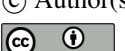



horizontal wind speed at 80 m above the surface. The time of occurrence of the first wind speed maximum varied between

11 and 15 hours following initialization, depending on model configuration and forcing.

For the convective case study, which requires much shorter spinup due to strong buoyant forcing dominating relevant

ABL characteristics, the model solutions were compared after one hour of spin-up time.

**1.6 Sensitivity experiments**

Sensitivities of the simulations to variability in model forcing, numerical methods, configuration, and turbulence SFS

models were obtained from a suite of simulations using different values of relevant parameters. Sensitivity to forcing was

examined by varying $U_g$ and $z_0$ around base values, estimated from wind profiler data and surface cover, respectively, and

using two representative values of $H_S$ during convective conditions. Configuration parameters included mesh resolution in

the vertical ($\Delta z$) and horizontal ($\Delta x = \Delta y$) directions, the combination of which determine two other parameters that impact

model performance, the grid aspect ratio, $\alpha = \Delta x / \Delta z$, and $l$. Vertical and horizontal mesh resolutions were therefore varied

independently to isolate sensitivities to each. Sensitivity to different orders of accuracy of advection schemes in both

horizontal, $O(h)$, and vertical $O(v)$, directions, was also examined.

While forcing and configuration parameters could be varied within all code bases, not all codes supported multiple

options for all parameters. The sensitivity experiments therefore involved changes both across and within the different codes.

Due to the large number of parameters, assessing the impacts of each independently was infeasible. Instead, forcing,

configuration, numerics and SFS turbulence options were combined into a large yet feasible suite of simulations listed in

Tables 1–2 (the * symbol in the SOWFA simulations indicates reductions of the model constants, to $C_S = 0.135$ and $C_e = 0.0673$, the latter resulting in an effective $C_S$ value of 0.135; see Sullivan et al., 1994). While results from all setups in

Tables 1– 2 were analyzed, for brevity only a subset is presented herein.



**Table 1.** Forcing and configuration parameters for the neutral-case sensitivity studies, using WRF (W#), SOWFA (S#) and HiGrad (H#), as described in the text.

|     | $Ug$ | $z0$ | $\Delta x$ | $\Delta z$ | $l$ | $\alpha$ | $nx$ | $nz$ | $O(h)$ | $O(v)$ | $SFS$ |
|-----|------|------|------|------|------|------|------|------|------|------|------|
| W1  | 6.5  | 0.05 | 25   | 7.5  | 16.74 | 3.3 | 96  | 176 | 5 | 3 | Lilly |
| W2  | 7.15 | 0.1  | 25   | 7.5  | 16.74 | 3.3 | 96  | 176 | 5 | 3 | Lilly |
| W3  | 5.85 | 0.01 | 25   | 7.5  | 16.74 | 3.3 | 96  | 176 | 5 | 3 | Lilly |
| W4  | 6.5  | 0.05 | 15   | 5    | 10.40 | 3   | 160 | 250 | 5 | 3 | Lilly |
| W5  | 6.5  | 0.05 | 25   | 7.5  | 16.74 | 3.3 | 96  | 176 | 2 | 2 | Lilly |
| W6  | 6.5  | 0.05 | 15   | 15   | 15.00 | 1   | 160 | 120 | 5 | 3 | Lilly |
| W7  | 6.5  | 0.05 | 25   | 7.5  | 16.74 | 3.3 | 96  | 176 | 2 | 2 | SMAG |
| W8  | 6.5  | 0.05 | 15   | 15   | 15.00 | 1   | 160 | 120 | 2 | 2 | SMAG |
| W9  | 6.5  | 0.05 | 25   | 7.5  | 16.74 | 3.3 | 96  | 176 | 5 | 3 | NBA-TKE |
| W10 | 7.15 | 0.1  | 25   | 7.5  | 16.74 | 3.3 | 96  | 176 | 5 | 3 | NBA-TKE |
| W11 | 5.85 | 0.01 | 25   | 7.5  | 16.74 | 3.3 | 96  | 176 | 5 | 3 | NBA-TKE |
| W12 | 6.5  | 0.05 | 15   | 5    | 10.40 | 3   | 160 | 250 | 5 | 3 | NBA-TKE |
| W13 | 6.5  | 0.05 | 25   | 7.5  | 16.74 | 3.3 | 96  | 176 | 2 | 2 | NBA-TKE |
| W14 | 6.5  | 0.05 | 15   | 15   | 15.00 | 1   | 160 | 120 | 5 | 3 | NBA-TKE |
| W15 | 6.5  | 0.05 | 25   | 7.5  | 16.74 | 3.3 | 96  | 176 | 2 | 2 | NBA-SR |
| W16 | 6.5  | 0.05 | 15   | 15   | 15.00 | 1   | 160 | 120 | 2 | 2 | NBA-SR |
| W17 | 6.5  | 0.1  | 25   | 7.5  | 16.74 | 3.3 | 96  | 176 | 5 | 3 | Lilly |
| W18 | 6.5  | 0.1  | 25   | 7.5  | 16.74 | 3.3 | 96  | 176 | 5 | 3 | SMAG |
| W19 | 6.5  | 0.1  | 25   | 7.5  | 16.74 | 3.3 | 96  | 176 | 5 | 3 | NBA-SR |
| S1  | 6.5  | 0.05 | 15   | 15   | 15.00 | 1   | 160 | 96  | 2 | 2 | Lilly |
| S2  | 7.15 | 0.1  | 15   | 15   | 15.00 | 1   | 160 | 96  | 2 | 2 | Lilly |
| S3  | 5.85 | 0.01 | 15   | 15   | 15.00 | 1   | 160 | 96  | 2 | 2 | Lilly |
| S4  | 6.5  | 0.05 | 7.5  | 7.5  | 7.50  | 1   | 320 | 175 | 2 | 2 | Lilly |
| S5  | 6.5  | 0.05 | 15   | 15   | 15.00 | 1   | 160 | 96  | 2 | 2 | Lilly |
| S6  | 6.5  | 0.05 | 25   | 7.5  | 16.74 | 3.3 | 96  | 175 | 2 | 2 | Lilly |
| S7  | 6.5  | 0.05 | 25   | 7.5  | 16.74 | 3.3 | 96  | 175 | 2 | 2 | SMAG |
| S8  | 6.5  | 0.05 | 15   | 15   | 15.00 | 1   | 160 | 96  | 2 | 2 | SMAG |
| S9  | 6.5  | 0.05 | 15   | 15   | 15.00 | 1   | 160 | 96  | 2 | 2 | SMAG* |
| S10 | 6.5  | 0.05 | 7.5  | 7.5  | 7.5   | 1   | 320 | 175 | 2 | 2 | Lilly* |
| S11 | 6.5  | 0.05 | 15   | 15   | 15.00 | 1   | 160 | 96  | 2 | 2 | NBA-TKE |
| S12 | 7.15 | 0.1  | 15   | 15   | 15.00 | 1   | 160 | 96  | 2 | 2 | NBA-TKE |
| S13 | 5.85 | 0.01 | 15   | 15   | 15.00 | 1   | 160 | 96  | 2 | 2 | NBA-TKE |
| S14 | 6.5  | 0.05 | 7.5  | 7.5  | 7.50  | 1   | 320 | 175 | 2 | 2 | NBA-TKE |
| S15 | 6.5  | 0.05 | 25   | 7.5  | 16.74 | 3.3 | 96  | 175 | 2 | 2 | NBA-TKE |
| H1  | 6.5  | 0.05 | 15   | 15   | 15.00 | 1   | 160 | 98  | 5 | 5 | Lilly |
| H2  | 7.15 | 0.1  | 15   | 15   | 15.00 | 1   | 160 | 98  | 5 | 5 | Lilly |
| H3  | 5.85 | 0.01 | 15   | 15   | 15.00 | 1   | 160 | 98  | 5 | 5 | Lilly |
| H4  | 6.5  | 0.05 | 7.5  | 7.5  | 7.50  | 1   | 320 | 174 | 5 | 5 | Lilly |
| H5  | 6.5  | 0.05 | 15   | 15   | 15.00 | 1   | 160 | 98  | 3 | 3 | Lilly |
| H6  | 6.5  | 0.05 | 25   | 7.5  | 16.74 | 3.3 | 96  | 174 | 5 | 5 | Lilly |

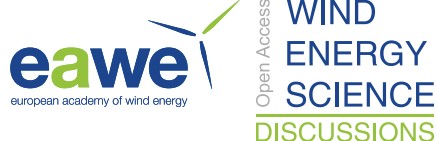

**Table 2.** Forcing and configuration parameters for the convective-case sensitivity studies, using WRF (W#), SOWFA (S#) and HiGrad (H#), as described in the text.

| | $Ug$ | $H_S$ | $\Delta x$ | $\Delta z$ | $l$ | $\alpha$ | $nx$ | $nz$ | $O(h)$ | $O(v)$ | $SFS$ |
|---|---|---|---|---|---|---|---|---|---|---|---|
| W1 | 9 | 0.3500 | 30 | 10 | 20.8 | 3 | 96 | 200 | 5 | 3 | Lilly |
| W2 | 9 | 0.3500 | 20 | 20 | 20 | 1 | 96 | 200 | 5 | 3 | Lilly |
| W3 | 9 | 0.3500 | 15 | 5 | 10.4 | 3 | 96 | 200 | 5 | 3 | Lilly |
| W4 | 10 | 0.4364 | 30 | 10 | 20.8 | 3 | 160 | 300 | 5 | 3 | Lilly |
| W5 | 10 | 0.3500 | 30 | 10 | 20.8 | 3 | 96 | 200 | 5 | 3 | Lilly |
| W6 | 9 | 0.3500 | 30 | 10 | 20.8 | 3 | 160 | 100 | 2 | 2 | SMAG |
| W7 | 9 | 0.3500 | 20 | 20 | 20.0 | 1 | 96 | 200 | 2 | 2 | SMAG |
| W8 | 9 | 0.3500 | 30 | 10 | 20.8 | 3 | 200 | 300 | 5 | 3 | NBA-TKE |
| W9 | 9 | 0.3500 | 20 | 20 | 20 | 1 | 300 | 200 | 5 | 3 | NBA-TKE |
| W10 | 9 | 0.3500 | 15 | 5 | 10.4 | 3 | 400 | 600 | 5 | 3 | NBA-TKE |
| W11 | 10 | 0.4364 | 30 | 10 | 20.8 | 3 | 200 | 300 | 5 | 3 | NBA-TKE |
| W12 | 10 | 0.3500 | 30 | 10 | 20.8 | 3 | 200 | 300 | 5 | 3 | NBA-TKE |
| W13 | 9 | 0.3500 | 30 | 10 | 20.8 | 3 | 200 | 300 | 2 | 2 | NBA-SR |
| W14 | 9 | 0.3500 | 20 | 20 | 20 | 1 | 300 | 150 | 2 | 2 | NBA-SR |
| S1 | 9 | 0.3500 | 20 | 20 | 20 | 1 | 300 | 150 | 2 | 2 | Lilly |
| S2 | 9 | 0.3500 | 10 | 10 | 10 | 1 | 600 | 300 | 2 | 2 | Lilly |
| S3 | 10 | 0.4364 | 20 | 20 | 20 | 1 | 300 | 150 | 2 | 2 | Lilly |
| S4 | 9 | 0.3500 | 30 | 10 | 20.8 | 3 | 200 | 300 | 2 | 2 | Lilly |
| S5 | 9 | 0.3500 | 20 | 20 | 20 | 1 | 300 | 150 | 2 | 2 | SMAG |
| S6 | 9 | 0.3500 | 20 | 20 | 20 | 1 | 300 | 150 | 2 | 2 | Lilly* |
| S7 | 9 | 0.3500 | 20 | 20 | 20 | 1 | 300 | 150 | 2 | 2 | SMAG* |
| S8 | 9 | 0.3500 | 10 | 10 | 10 | 1 | 600 | 300 | 2 | 2 | Lilly* |
| S9 | 9 | 0.3500 | 20 | 20 | 20 | 1 | 300 | 150 | 2 | 2 | NBA-TKE |
| S10 | 9 | 0.3500 | 10 | 10 | 10.0 | 1 | 600 | 300 | 2 | 2 | NBA-TKE |
| S11 | 10 | 0.4364 | 20 | 20 | 20 | 1 | 300 | 150 | 2 | 2 | NBA-TKE |
| S12 | 9 | 0.3500 | 30 | 10 | 20.8 | 3 | 200 | 300 | 2 | 2 | NBA-TKE |
| H1 | 9 | 0.3500 | 20 | 20 | 20 | 1 | 96 | 200 | 5 | 5 | Lilly |
| H2 | 9 | 0.3500 | 10 | 10 | 10 | 1 | 96 | 200 | 5 | 5 | Lilly |
| H3 | 10 | 0.4364 | 20 | 20 | 20 | 1 | 96 | 200 | 5 | 5 | Lilly |
| H4 | 9 | 0.3500 | 10 | 10 | 10 | 1 | 160 | 300 | 3 | 3 | Lilly |
| H5 | 9 | 0.3500 | 30 | 10 | 20.8 | 3 | 96 | 200 | 5 | 5 | Lilly |

**3 Results**

5 **3.1 Qualitative assessment**

First, high level results from the sensitivity simulations are shown to indicate some key differences between the case studies and solvers. A more detailed comparison of various flow parameters from the simulations is provided in Sect. 3.2.

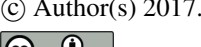


### 3.1.1 Neutral case

Figure 1 shows an instantaneous horizontal wind speed field both in both plan $x$-$z$ cross-section at 100 m above the surface (top panels) and in a vertical $x$-$z$ cross-section at the $y$-direction midpoint, from all three solver bases. The same forcing is used for all simulations, with the exception being the geostrophic wind direction, which was set to westerly in the

5   HiGrad simulations, rather than northwesterly in WRF and SOWFA. Due to the idealized setup, using flat terrain, homogeneous surface and atmospheric forcing, and periodic LBCs, the only effect of this difference is to rotate the simulated wind direction, with no impacts on relevant flow characteristics. Each simulation used grid resolution of 15 m in all directions, with the discretization of the advective term the lowest order option, $O(h) = O(v) = 2$, in WRF-LES and SOWFA, and 3 in HiGrad.

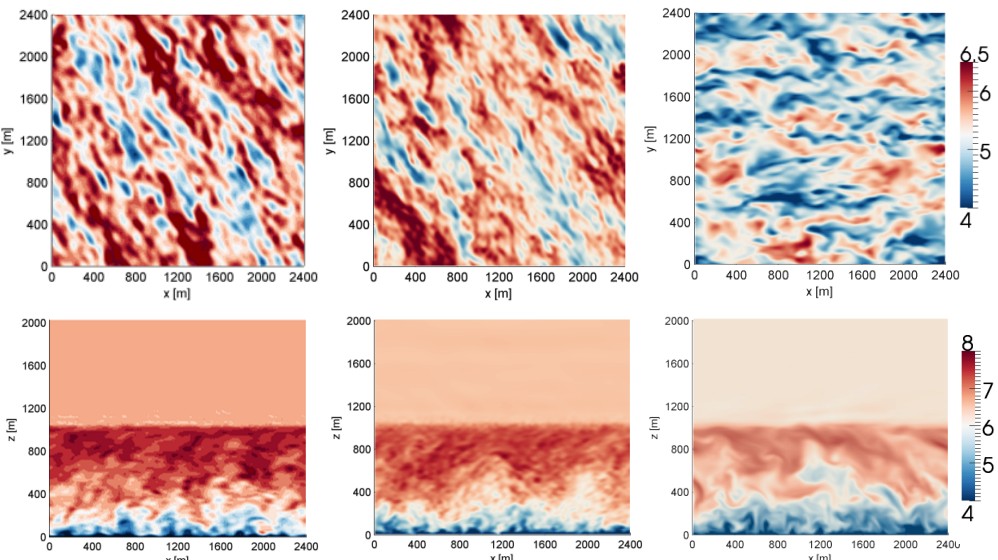

**Figure 1: Comparisons of instantaneous horizontal wind speed [m s$^{-1}$] from the neutral case study, at 100 m above the surface (top row) and in a cross-stream plane midway through the domain (bottom row) from WRF (left), SOWFA (middle) and HiGrad (right). In each case, the external forcing was the same, grid resolution was 15 m in each direction, and the advective scheme was the lowest-order option for each solver.**

15   While differences among the solutions are apparent, all three solvers show similar characteristic turbulence structures, namely the elongated low-speed streamwise structures, a range of sizes of turbulence structures, diminishing with increasing proximity to the surface, and similar ABL heights, due to the capping inversion.

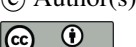



Figure 2 shows the impacts of increasing both the grid resolution and the accuracy of the advective operators within the same solver, in this case HiGrad, on instantaneous wind speed, in the same two planes as Fig. 1. The grid spacing was decreased by a factor of two in all directions, while $O(h)$ and $O(v)$ were increased from 3 to 5. Results of these changes include a broader range of flow structures, particularly at the small-scale end of the spectrum, due to more of the inertial

subrange being explicitly resolved, as well as the wider range of magnitudes, with both lower minima and higher maxima within the resolved structures. Similar impacts were observed for all three solvers under corresponding changes to grid resolution and advection schemes (not shown).

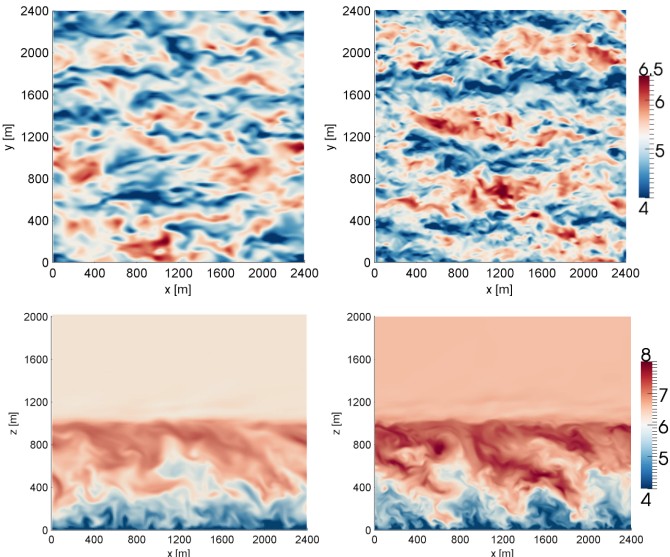

**Figure 2: Comparisons of instantaneous horizontal wind speed [m s⁻¹] from the neutral case study, at 100 m above the surface (top**
**row) and in a cross-stream plane midway through the domain (bottom row) from the HiGrad solver operating at 15 m resolution in each direction and using order 3 advective discretization (left), and 7.5 m resolution in each direction and order 5 advective discretization (right).**

### 3.1.2 Convective case

Figure 3 shows instantaneous cross sections of potential temperature in both the $x$-$y$ plane at 100 m above the surface

(top), and the $x$-$z$ plane at the domain $y$-direction midpoint (bottom), from the convective case study, using the WRF (left), SOWFA (middle) and HiGrad (right) solvers, as in Fig. 1. Each of the simulations shown in Fig. 3 used identical physical forcing ($U_g$, $H_S$, and $z_0$,), however different numerical settings and grid configurations were employed. With near-surface

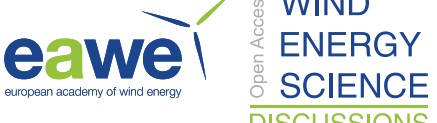



flow parameters showing strong sensitivity to the value of $\alpha$ near the surface in previous LES studies (Brasseur and Wei,

2010; Mirocha et al., 2010), each simulation shown in Fig. 3 utilized the $\alpha$ value that produced the best agreement with the

expected logarithmic similarity solution in the surface layer within that solver base; $\alpha \cong 1 - 2$ for SOWFA and HiGrad, and

$\alpha \cong 3 - 4$ for WRF. Therefore, the HiGrad and SOWFA simulations shown in Fig. 3 use an isotropic grid with $\alpha = 1$ and

with grid cell sizes of 20 m in each direction, while WRF uses $\alpha = 3$, with horizontal and vertical grid spacings of 30 and 10

m, respectively. To compensate for the coarser horizontal resolution, the WRF simulation used its highest-order advection

options, $O(h) = 5$ and, $O(h) = 3$, while the lowest-order options, 2 and 3, were used for SOWFA and HiGrad, respectively.

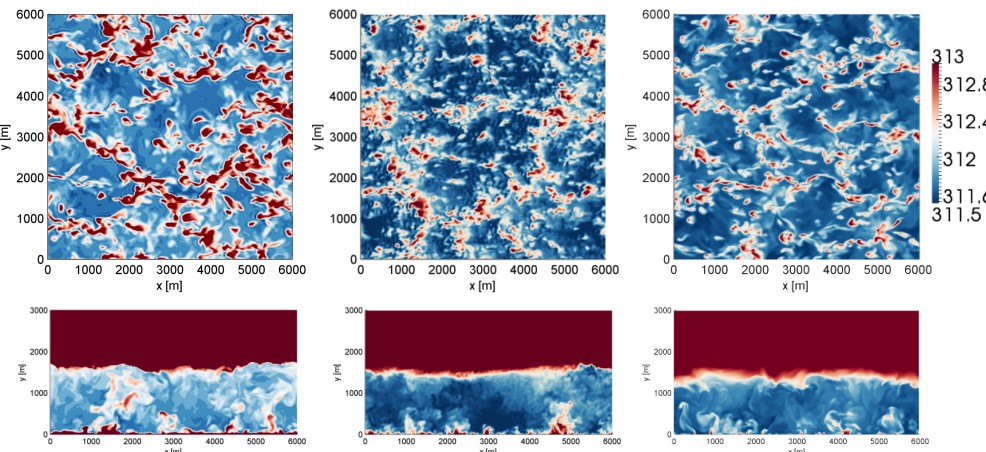

**Figure 3: Comparisons of instantaneous potential temperature [K] from the unstable case study, at 100 m above the surface (top**
**row) and in a cross-stream plane midway through the domain (bottom row) from WRF (left), SOWFA (middle) and HiGrad**
**(right). Each simulation used the same external forcing but different solver options, as described in the text.**

Inspection of Fig. 3 reveals qualitative similarities in resolved flow characteristics, including the shapes and sizes of the

turbulent structures in both cross sections. However, the WRF simulations exhibit less fine-scale structure than the others,

despite the use of higher resolution in the vertical direction, and higher-order advection operators, indicating that horizontal

resolution is the dominant factor influencing the size distribution of resolved scales, within the examined range of parameter

values. The slightly higher temperatures within the WRF ABL (Fig. 3) are most likely artifacts of the Rayleigh damping

imposed above the ABL, which relaxes temperature back to its initial value beginning just above the ABL top.





Figure 4 isolates the impact of changing only the mesh resolution (by a factor of three in each direction) within the same solver (WRF) while leaving all other settings constant. Instantaneous cross sections in the same two planes as shown in Fig. 4, from the coarse- (left) and fine-resolution (right) simulations, show that while both resolutions capture the same morphological characteristic, most notably quasi-cellular convective cells of similar sizes and magnitudes, an increased

range of scales of motion are captured with the finer-resolution LES.

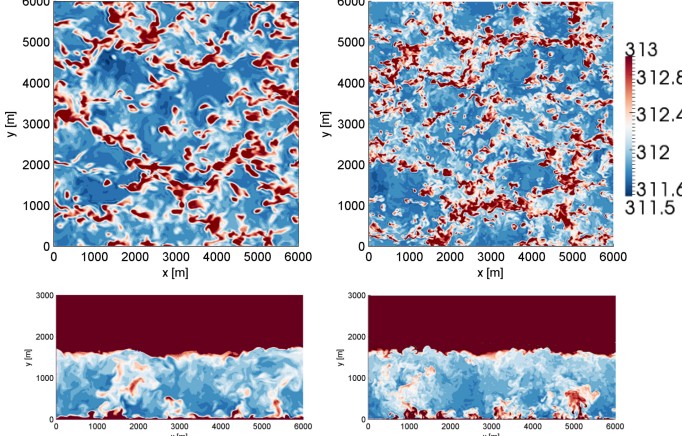

**Figure 4: Comparison of instantaneous potential temperature in the convective case at 100 m above the surface (top row) and in a cross-stream plane midway through the domain (bottom row) from the WRF-LES solver operating at 30 m horizontal by10 m vertical resolution (left) and 15 m horizontal by 5 m vertical resolution (right).**

**3.2 Quantitative assessment**

The ABLs and simulations thereof comprising this study are approximately horizontally homogeneous and therefore averaging over horizontal planes could be applied for assessment. However, considering that future planned studies will involve heterogeneous boundary layers under time-varying forcing, temporal averaging and spectral analysis in the frequency domain is instead utilized. Simulation results therefore consist of a single vertical profile located near the center of

the computational domain, output every second (1 Hz) during the time window of analysis.

**3.2.1 Neutral case**

As described above, the evening transition of August 17, 2012 provided the best approximation to canonical near neutral ABL conditions within the observational dataset, however subsequent detailed analysis of turbulent kinetic energy (TKE)

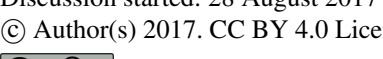



measured with sonic anemometers at the SWiFT tower showed that sonic anemometers were partially in the wake of the tower, resulting in larger measured TKE values than what would be expected in unobstructed flow under the same conditions. As the tower cross section and lattice structure comprise length scales much smaller than the characteristic production scales of turbulence for the considered ABL types, most of the covariance, arising mostly from the largest eddies,

is hypothesized to have been only minimally impacted by the tower. Therefore, while preventing a detailed comparison of TKE, other parameters not strongly impacted by the quasi-random perturbations created by tower interactions, such as turbulent stresses and velocity spectra and cospectra, were compared qualitatively. Mean wind speed and direction profiles, which showed no evidence of tower wake influence, were compared quantitatively, as described below.

### Sensitivity to model configuration

Figure 5 shows time averaged profiles of wind speed from simulations using all three solver bases, compared both against measurements at the SWiFT tower (left), and to the theoretical logarithmic profiles in the surface layer (right). Measurement variability is shown as "uncertainty" bars that signify one standard deviation from a mean value computed as a 90-minute time average. All simulations use the Lilly SFS model and the highest-order advection option available. Two different grid setups were used, with horizontal and vertical grid sizes of 25 and 7.5 m for WRF, resulting in $\alpha = 3$, and 15

m each for SOWFA and HiGrad, resulting in $\alpha = 1$, yielding optimal performance for each model, as described in Sect. 1.6.

Despite the use of different numerical and grid specifications, all simulations produced generally good agreement with measurements, falling within measurement variability. The measured wind speed profile does not increase monotonically with height, as would be expected in canonical ABL flow, indicating the presence of height-dependent transient processes and forcings. Considering that such processes cannot be captured with idealized forcing and simulation setups, the agreement

between model output and the data can be considered quite good. The logarithmic profile in the surface layer is also captured well, despite the known tendency of the Lilly SFS parameterization to overpredict non-dimensional shear relative to a logarithmic profile in the surface layer of a neutral ABL (Brasseur and Wei, 2010; Mirocha et al., 2010).

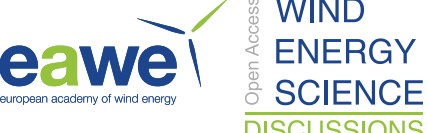



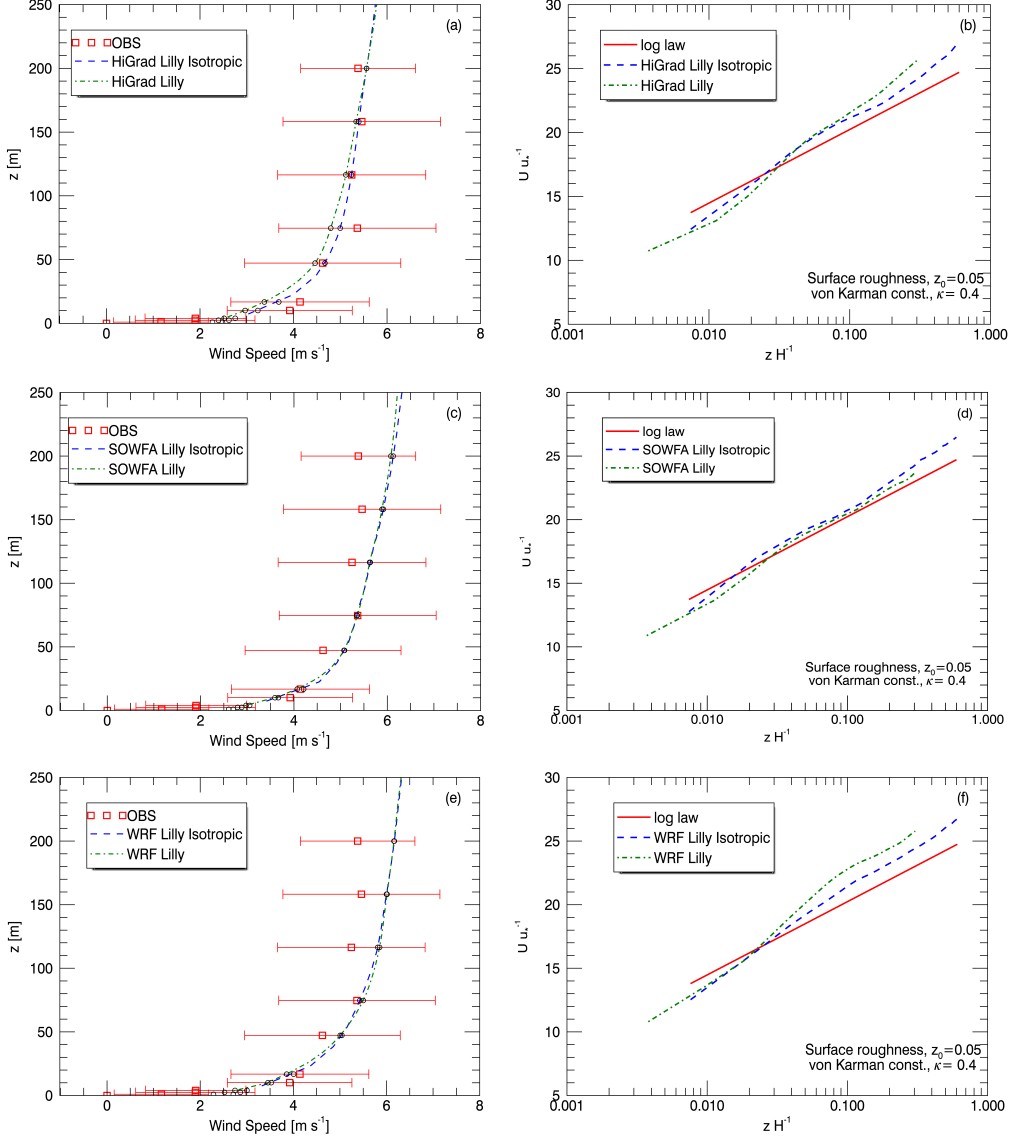

**Figure 5: Simulated profiles of time averaged wind speed plotted against observations (left: a, c, e) and the theoretical logarithmic profile shape (right: b, d, f), from the neutral case study, using HiGrad (top: a, b), SOWFA (middle: c, d), and WRF (bottom: e, f).**

5   Figure 6 compares simulated wind speed profiles using the three models, all with an isotropic grid formulation, while

also showing the impact of using two different SFS parameterizations in WRF, Lilly and NBA-TKE. Again, results are



generally similar, with HiGrad showing slightly smaller wind speeds above 50 m, agreeing slightly better with the mean of

the observations, than SOWFA and WRF, with all models accurately reproducing near-surface shear, displaying good

agreement with the logarithmic profile.

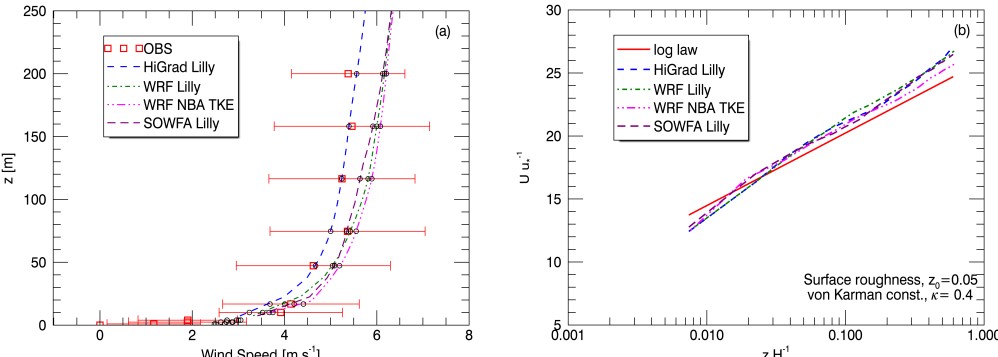

**Figure 6: Impacts of different solvers, all using isotropic grids of 15 m in each direction, on simulated profiles of time averaged wind speed plotted against observations (a) and the theoretical logarithmic profile shape (b), from the neutral case study.**

The impact of different advection operators was also analyzed. Here, only results from HiGrad and WRF are presented,

as SOWFA includes only one advection option. Figure 7 shows results from HiGrad with $O(h) = O(v) = 3$ and 5 upwind

advection schemes, indicating that better agreement with measurements is obtained using higher-order schemes. Figure 8

shows the impact of different combinations of advective scheme and SFS stress model on WRF's wind speed profiles,

indicating that, for this suite of simulations, either configuration choice results in variability of similar magnitude.

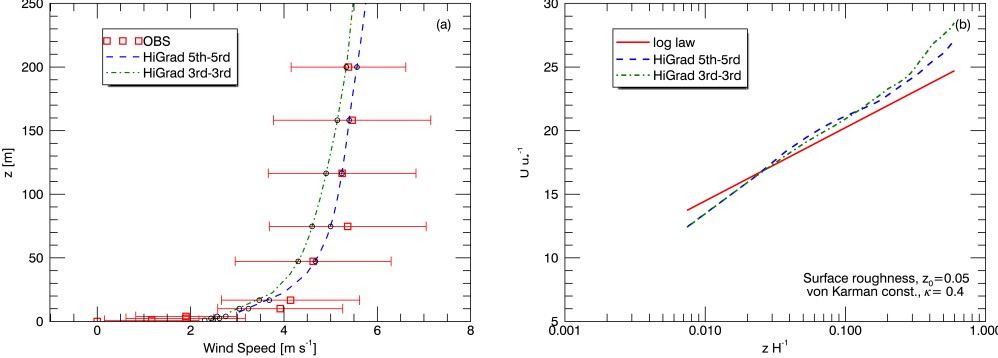

**Figure 7: Impacts of different advection schemes within the HiGrad model, on simulated profiles of time averaged wind speed**
**plotted against observations (a) and the theoretical logarithmic profile shape (b), from the neutral case study.**

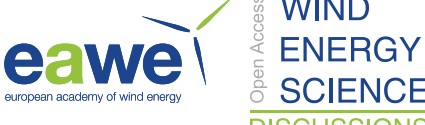

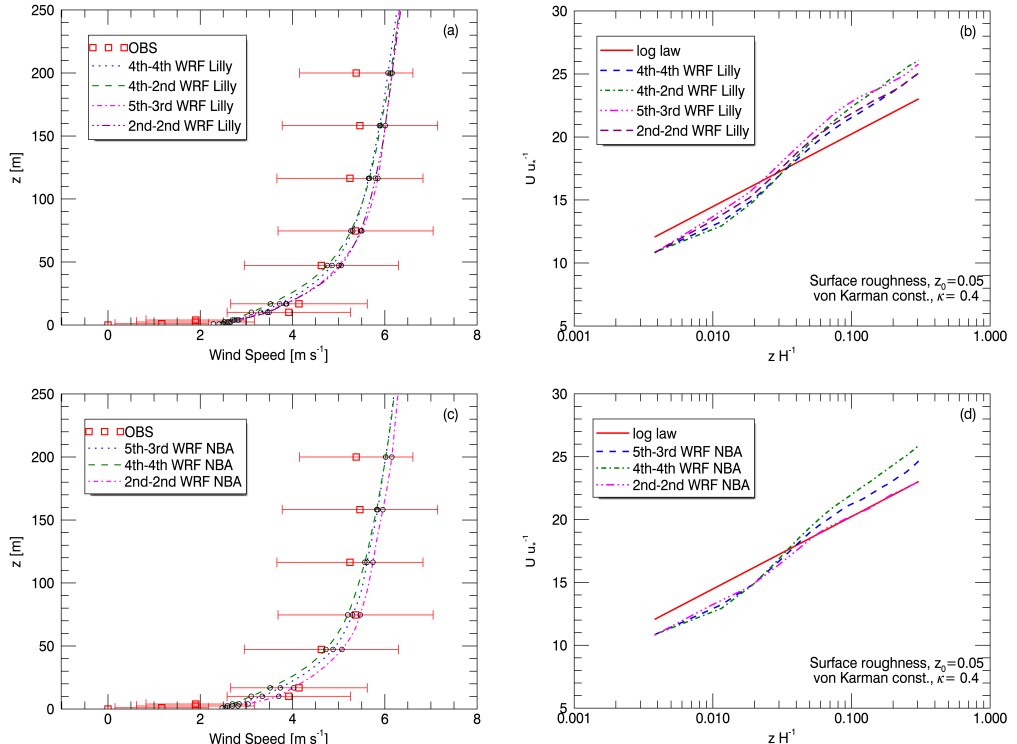

**Figure 8:** Impacts of varying advection schemes and SFS models in WRF, on simulated profiles of time averaged wind speed plotted against observations (left: a, c) and the theoretical logarithmic profile shape (right: b, d), from the neutral case study. Simulations used Lilly (top: a, b) and NBA (bottom: c, d) SFS parameterizations.

5        The relative performances of various configurations are assessed quantitatively using the Mean Absolute Error (MAE) and Root Mean Square Error (RMSE), and MAE and vertical shear across a representative turbine rotor (the range of heights spanned by the turbine blades), relevant quantities for both power production and fatigue loading. MAE and RMSE values were computed by interpolating LES results to measurement levels. Vertical shear MAE is computed as a difference between

10      predicted and measured shear at all measurement levels across a 100 m rotor with a hub height of 90 m. Table 3 shows values of each parameter, indicating that simulations using the higher-order advections generally capture more accurately velocity profile across both the tower and the rotor.





**Table 3. Analysis of HiGrad performance using different advection schemes with the Lilly SFS parameterization.**

| O(h) | O(v) | Tower MAE | Tower RMSE | Rotor MAE | Rotor Shear MAE |
|------|------|-----------|------------|-----------|-----------------|
| 5 | 5 | 0.469137 | 0.618180 | 0.962881 | 0.464641 |
| 3 | 3 | 0.567594 | 0.643881 | 0.780441 | 0.469093 |

Similar analysis was performed using the WRF model varying the order of the advection scheme and the SFS parameterizations, as summarized in Tables 4 and 5. In this study, $O(h) = 3$, 4, and 5 were used in horizontal, and $O(v) = 2$

and 3 in the vertical directions. These advection schemes were varied in combination with the Lilly, NBA-SR and NBA-TKE SFS parameterizations. Tables 4 and 5 show that using different advection schemes results in variability of comparable or even greater magnitude than that resulting from different SFS parameterizations. In general, even-order schemes perform better than upwind schemes (Table 4 and Table 5), perhaps due to even-order schemes being less dissipative. However, when the NBA SFS parameterization is used, all advection options produce very similar results, with a slight advantage

(Table 5) of the odd-order scheme when using the NBA options.

As numerical simulations of homogeneous boundary layers generally represent ideal conditions, more realistic simulations may include significant spatial gradients associated with, for example, microfronts. For such applications, odd-order upwind schemes would likely be advantageous. The analysis of WRF results indicates that the choice of the advection scheme could be as important as the choice of a subfilter stress parameterization and that the best performance is obtained

with specific combinations of SFS parameterizations and advection schemes (also see Fig. 8).

**Table 4. Analysis of WRF LES performance using different advection schemes with the Lilly SFS parameterization.**

| O(h) | O(v) | Tower MAE | Tower RMSE | Rotor MAE | Rotor Shear MAE |
|------|------|-----------|------------|-----------|-----------------|
| 4 | 4 | 0.599974 | 0.755268 | 1.22615 | 0.235284 |
| 4 | 2 | 0.549931 | 0.669788 | 1.04795 | 0.235554 |
| 5 | 3 | 0.697584 | 0.837003 | 1.30949 | 0.235102 |
| 3 | 3 | 0.71092 | 0.868417 | 1.38102 | 0.235457 |




**Table 5. Analysis of WRF LES performance using different advection schemes with the NBA and NBA-TKE SFS parameterization**.

| SFS | $O(h)$ | $O(v)$ | Tower MAE | Tower RMSE | Rotor MAE | Rotor Shear MAE |
|---|---|---|---|---|---|---|
| NBA | 4 | 4 | 0.580661 | 0.683801 | 1.03934 | 0.234796 |
| NBA | 4 | 2 | 0.585018 | 0.680622 | 1.00501 | 0.234952 |
| NBA | 5 | 3 | 0.512753 | 0.610661 | 0.937374 | 0.233994 |
| NBA-TKE | 4 | 4 | 0.607639 | 0.695919 | 0.935463 | 0.236384 |
| NBA-TKE | 4 | 2 | 0.582866 | 0.675690 | 0.931736 | 0.236381 |
| NBA-TKE | 5 | 3 | 0.767281 | 0.907598 | 1.37395 | 0.235458 |

### Sensitivity to forcing parameters

Assessment of sensitivity to two key boundary conditions and forcing parameters, $z_0$ and $U_g$, is also performed. Each of these parameters is typically held uniform in space and constant in time in idealized LES, and therefore must represent average values. The baseline values for these parameters, $z_0 = 0.05$ m and $U_{g,0} = 9$ m s$^{-1}$, were bracketed by two additional

10 cases, $z_0 = 0.01$ m and $U_g = 0.9U_{g,0}$, and $z_0 = 0.1$ m and $U_g = 1.1U_{g,0}$. The wind speed profiles resulting from changes of these parameters in all three models are shown in Fig. 9, indicating that each model exhibits the expected behavior of increasing wind speed at upper measurement levels of the SWiFT tower when $U_g$ is increased (Fig. 9, left panels). The WRF model shows the greatest sensitivity to changes of these parameters, with HiGrad showing the least. The effects of varying $z_0$ are better seen by comparing surface layer wind speed profiles with the logarithmic profiles shown in panels on the right

15 of Fig. 9. While all profiles appear nearly logarithmic, each model generates a slightly different slope of the wind speed profile near the surface, with simulations using the smaller $z_0$ values showing smaller departures from the baseline profiles than those using increased values, due to the $z_0$ value being a factor of 5 smaller, versus only a factor of 2 larger, in the reduced and increased value cases, respectively.

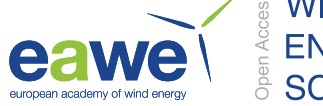

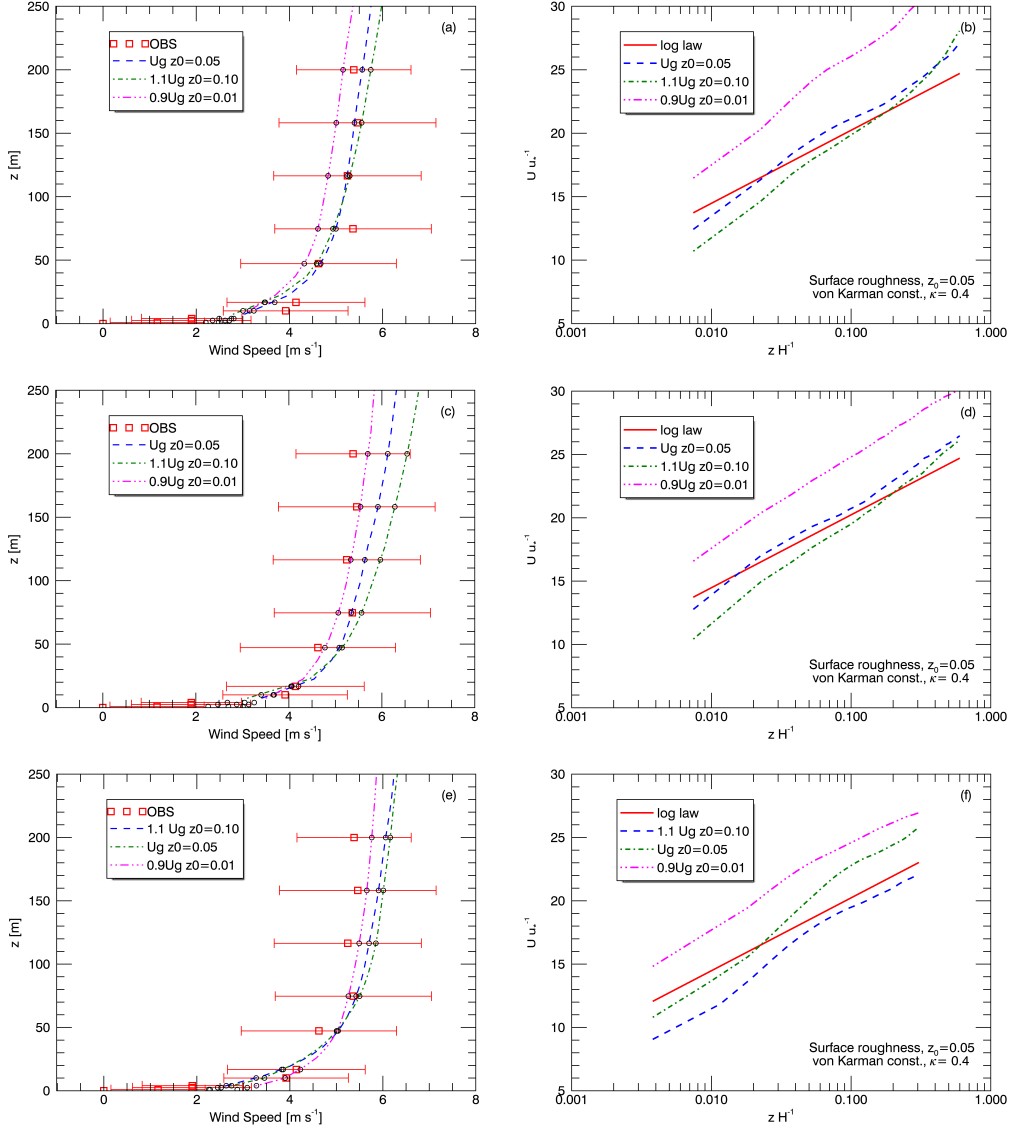

**Figure 9: Impacts of varying surface roughness length and geostrophic wind speed using HiGrad (top: a, b), SOWFA (middle: c, d) and WRF (bottom: e, f), on simulated profiles of time averaged wind speed plotted against observations (left: a, c, e) and the theoretical logarithmic profile shape (right: b, d, f), from the neutral case study.**

Differences in response to varying surface boundary conditions among the models can likely be attributed to differences

in implementation of surface boundary conditions. Considering the infeasibility of resolving the viscous sublayer of a high-

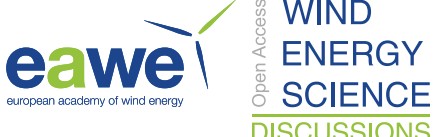

Reynolds number ABL flow, due to both extreme computational demands and uncertainties in details of terrain and surface cover, LES of ABLs generally rely on approximate surface boundary conditions that are in some form based upon the assumption of a developed logarithmic surface layer profile, modified by atmospheric stability (e.g., Moeng, 1984).

**Sensitivity to model grid resolution**

The preceding analysis of sensitivity to model configuration and forcing parameters utilized simulations conducted with moderately-fine grid resolutions. A more detailed assessment of model performance based on higher-resolution simulations of the baseline case of $z_0 = 0.05$ m and $U_{g,0} = 9$ m s$^{-1}$ was also conducted.  Each of the higher-resolution simulations was configured according each model's optimal $\alpha$ value, with HiGrad and SOWFA using an isotropic grid with $\alpha = 1$, with grid cell sizes of 7.5 m in each direction, while WRF used $\alpha = 3$, with horizontal and vertical grid spacings of 15 and 5 m,

respectively. To maintain the same domain size, HiGrad and SOWFA used 320×320×200 grid cells in the $x$-, $y$- and $z$-directions, while WRF used 160×160×300, respectively. All simulations used the Lilly SFS model.

Comparison of simulated and observed time-averaged wind speed profiles is shown in Fig. 10, top left panel. Excellent agreement is observed between SOWFA and WRF model results, each predicting slightly higher magnitudes than HiGrad, with all simulations falling within the range of the observation variability. Each simulation also produced good agreement

with the logarithmic profile in the surface layer (top right panel in Fig. 10). Temporal variability of the ten-minute average wind speed value for each simulation is shown in the other panels of Fig. 10, denoted as "error" bars, representing one standard deviation from the mean over all ten-minute averages. All three models result in similar temporal variability, all markedly lower than that of measured profiles. The difference between simulated and measured variability could be attributed to the fact that idealized simulations forced with constant and uniform $U_g$ did not account for possible variability

in large scale forcing that could be associated with the evening transition. Also, the SWiFT tower wake effects, as described in Sect. 3.2.1, have likely artificially enhanced velocity variations, as is discussed in more detail below. Moreover, simulated variability was calculated using only resolved velocity fluctuations, ignoring the SFS component which may have further increased the range of variability from the simulations.





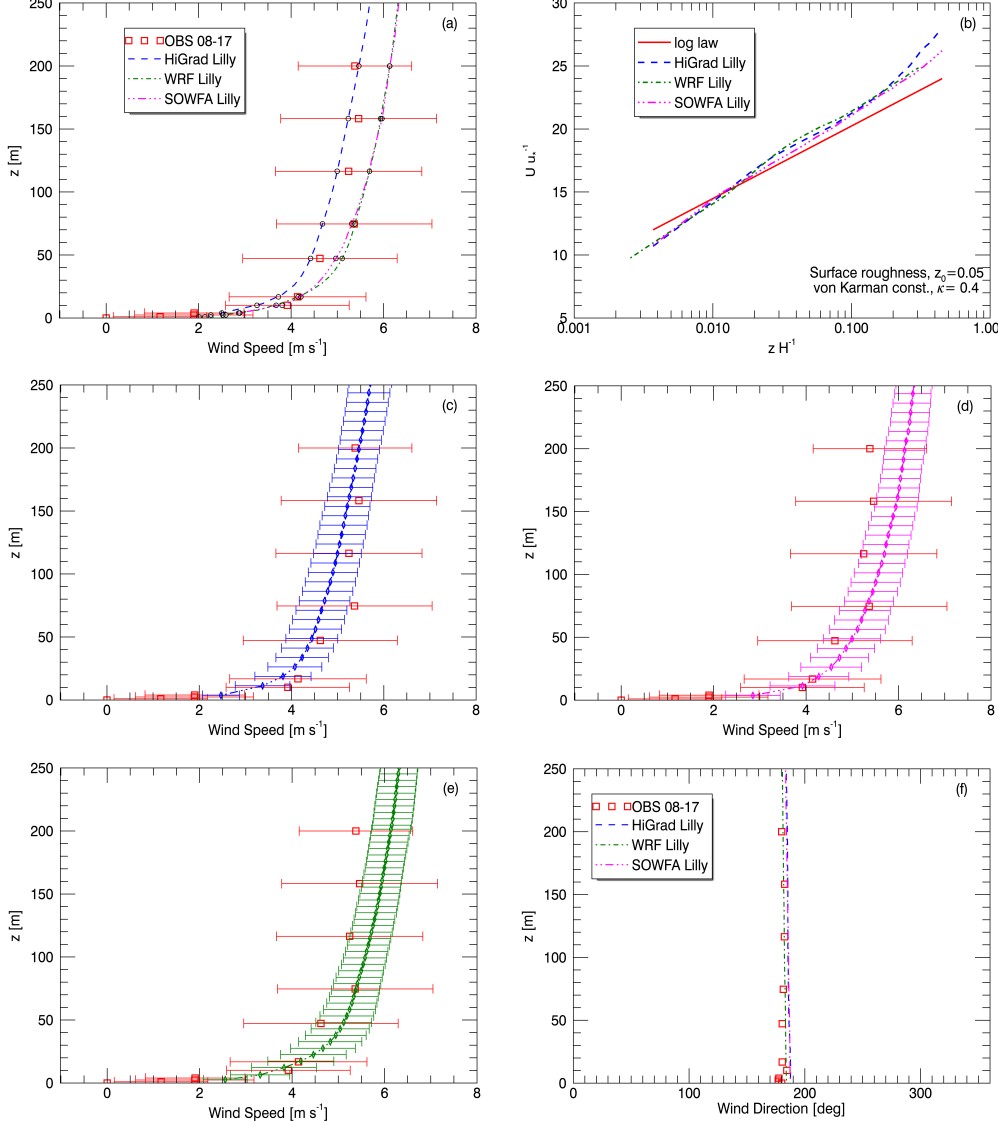

**Figure 10: Observed and simulated wind speed and direction, from the neutral case study, from each solver using its optimal aspect ratio and the Lilly SFS model. Top panels show each models' mean wind speeds against observed variability (left) and the theoretical logarithmic distribution (right). Middle and lower panels show each model's mean and variability relative to the observations, with wind direction shown in the lower right.**





For completeness, comparison between measured and simulated wind direction is shown in the bottom right panel of Fig. 10. Excellent agreement is observed except for a small difference at the lowest levels that could be potentially attributed to the effects of small terrain heterogeneities not represented within the simulations.

Quantitative MAE and RMSE values from the high-resolution simulations are presented in Table 6. While all the

models perform well, the HiGrad simulations produce lowest values of wind speed MAE and RMSE over all ten level measurements on the SWiFT tower, as well as the lowest value of wind speed MAE over a turbine rotor disk. The SOWFA simulations achieve the lowest shear MAE over a turbine rotor disk. It is noted that configurations used for the high-resolution simulations may not have been optimal for each model. As discussed earlier, using model configurations with different combinations of SFS models and advection schemes may yield better performance. In addition, uncertainty in the

forcing conditions and the unsteadiness of the evening transition ABL may have contributed to these errors. Finally, WRF's relatively lower scores are likely partially attributable to use of a factor of two coarser horizontal resolution (relative to the other models), a key modulator of resolved turbulence scales.

**Table 6. Analysis of high-resolution LES performance using different models with the Lilly SFS parameterization for the neutrally stratified ABL observed on August 17, 2012.**

|        | Tower MAE | Tower RMSE | Rotor MAE | Rotor Shear MAE |
|--------|-----------|------------|-----------|-----------------|
| HiGrad | 0.432486  | 0.493493   | 0.602712  | 0.242731        |
| WRF    | 0.632616  | 0.797163   | 1.28059   | 0.285634        |
| SOWFA  | 0.507027  | 0.618256   | 0.925746  | 0.108117        |

In addition to wind speed and direction, time-averaged profiles of vertical turbulent stresses, $u_* = [(\tau_{13})^2 + (\tau_{23})^2]^{1/4}$ with $\tau_{13} = \langle u'w' \rangle$ and $\tau_{23} = \langle v'w' \rangle$, and TKE $= 0.5(\langle u'u' \rangle + \langle v'v' \rangle + \langle w'w' \rangle)$, were also examined. Here $a'$ represents an instantaneous deviation from an average value, with $\langle a \rangle$ representing the averaging. Measured values of stresses and TKE were computed from tilt corrected and detrended high-rate (50 Hz) measurements, while simulated values used 1 Hz output,

and include both resolved and SFS components. Measurements were subsampled to 1 Hz to match the simulation output.

Figure 11 shows time-averaged turbulent stresses (left) and TKE (right) from both simulations and observations. All quantities were computed over a 90 minute period using 15 minute running values. In addition to the August 04 case,

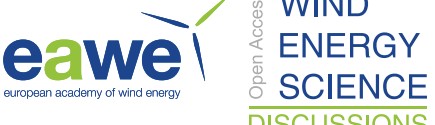

observations here include an additional near-neutral period occurring during the morning of July 10, 2012, which featured

similar but slightly greater wind speeds (by approximately 1 m s$^{-1}$ over the depth of the tower), but from a different direction

that avoided tower wake contamination. Agreement between the magnitudes of the simulated and observed stress is

generally good, with similar values observed during both periods. Simulated TKE values, however, are significantly smaller

than observed values during both periods, with observed magnitudes also differing substantially between the two periods.

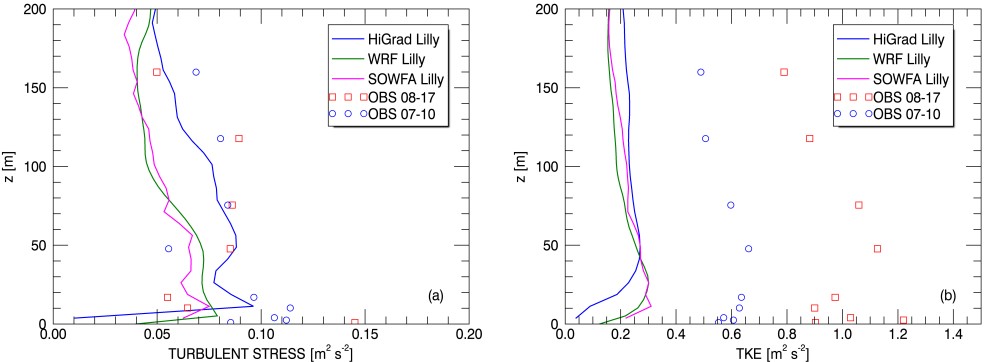

**Figure 11: Simulated and measured turbulent stress (a) and TKE (b), from the neutral case study.**

An explanation for the large differences between measured and observed TKE values, despite similar stress values, is

that tower wake effects likely contributed small, uncorrelated perturbations, enhancing the variances contributing to TKE

while not strongly impacting the covariances that determine the stress. The larger observed TKE values during the unwaked

July 10 case are likely due to greater vertical wind shear occurring within the stable conditions preceding the near neutral

morning transition period. Observed TKE values during the August 17 case also could have been influenced by residual

turbulence from the previous afternoon's convection. These factors highlight difficulties inherent in comparing observations

taken during near-neutral periods within a diurnal cycle, to idealized neutral simulations forced with no diurnal variability.

Despite the omission of diurnal variability (and other simplifications) in the idealized setups used herein, the stress values,

critical factors in turbine fatigue loading, were well captured.

Figure 12 shows power spectra of streamwise (top left) and vertical velocity (top right) components, as well as cospectra

of two turbulent stress components $\tau_{13} = \langle u'w' \rangle$ and $\tau_{23} = \langle v'w' \rangle$. All spectra and cospectra are computed at a

representative wind turbine hub height between 80 and 90 m, but at slightly different heights due to differences between the

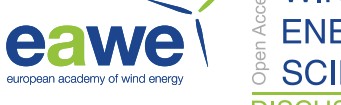



grid-cell height values of the simulations and the tower instrumentation. Values were computed from the 1 Hz data and model output by dividing a 90-minute time series into overlapping fifteen-minute intervals (overlapping over seven and a half minutes) and averaging the resulting eleven spectra and cospectra.

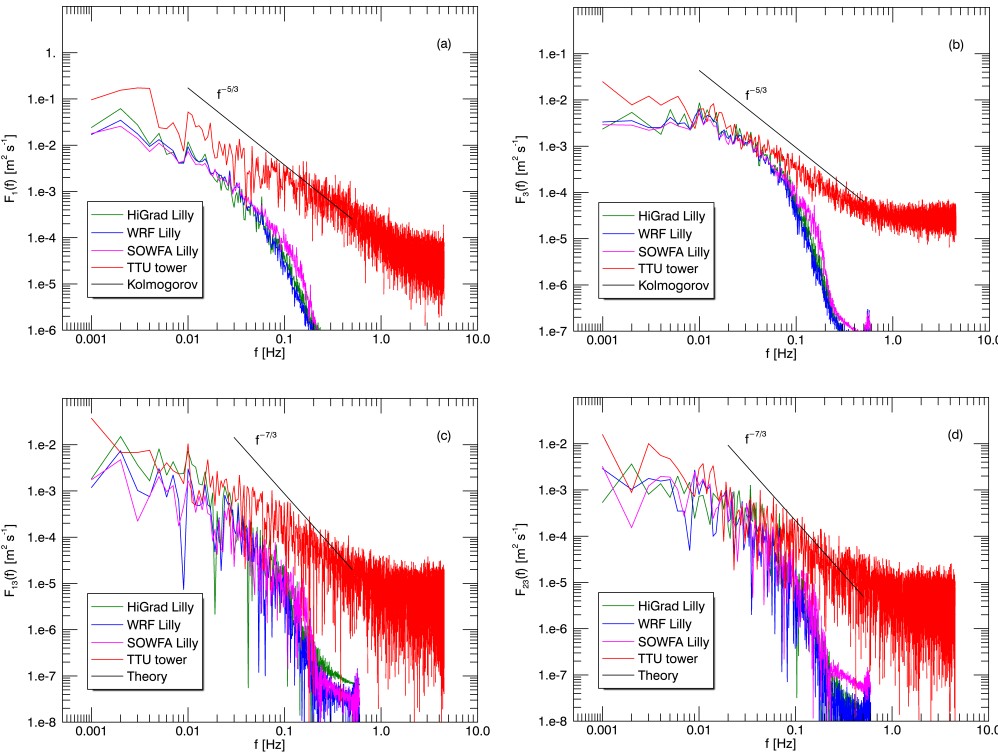

5   **Figure 12: Simulated and measured spectra of the streamwise (a) and vertical velocity (b) velocity, and cospectral components $Co\{u'w'\}$ (c) and $Co\{v'w'\}$ (d), from the neutral August 17 ABL case, at 80-90 m above the surface.**

The spectra shown in Fig. 12 (top) suggest that the primary cause of the larger measured than simulated TKE values is increased variability in the observed horizontal velocity components relative to the simulations, likely due to tower wake

10   effects. In contrast, the cospectra (bottom) show much better agreement between simulations and observations due to their dependence upon correlated structures produced by nonlinear dynamics, rather than the generally uncorrelated structures produced by the lattice tower.





Spectra and cospectra computed from model output display a high-wave-number drop-off characteristic of finite

difference and finite volume discretization schemes. A numerical scheme without full spectral resolution acts as a low-pass

filter, where the filter width depends on the type and order of the numerical scheme (Kosović et al., 2002; Skamarock, 2004).

As can be seen from Fig. 12, all three models exhibit similar high-wave-number drop-off characteristics, as expected, with

5    the SOWFA results producing slightly wider inertial subranges due to use of an even-order, centered scheme.

Figure 13 shows velocity spectra and cospectra, as in Fig. 12, from the unwaked July 10 case. As with the August 17

case, observed and simulated cospectra and vertical velocity spectra again agree well with each other. However, the absence

of spurious, tower-induced horizontal velocity perturbations greatly improves agreement between the measured and

simulated horizontal velocity spectra.

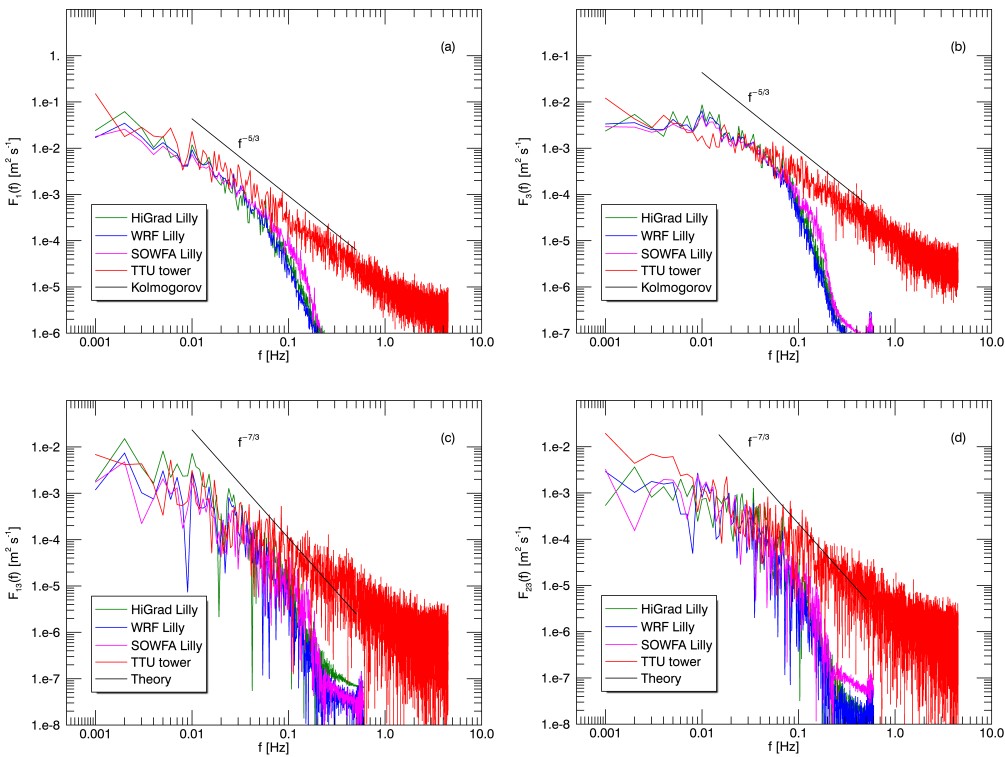

**Figure 13: Simulated and measured spectra and cospectra, as in Fig. 12, from another neutral case study occurring July 10, 2012, exhibiting no tower wake effects.**

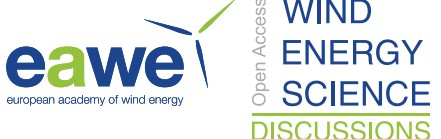

### 3.2.2 Convective case

Simulations during convective conditions were assessed using many of the same criteria applied to the neutral simulations, again evaluated using each solver's optimal $\alpha$ values, here using coarser resolution than the neutral case, with HiGrad and SOWFA using grid cell sizes of 20 m in each direction, while WRF used grid spacings of 30 and 10 m,

5   respectively. All simulations were again configured with the Lilly SFS model. HiGrad and WRF used high-order upwind schemes, with HiGrad using $O(h) = O(v) = 5$, while WRF used $O(h) = 5$ and $O(v) = 3$. SOWFA used $O(h) = O(v) = 2$.

Figure 14 shows wind speed (left) and direction (right) profiles, again with "measurement variability" bars on the wind speeds indicating one standard deviation about the mean of the ten-minute average values from the observations, from each solver. During convective conditions, the WRF results show the closest agreement with the mean values of the observed

10   wind speed, with SOWFA and HiGrad producing slightly slower values, with all predictions falling within the range of observed values.

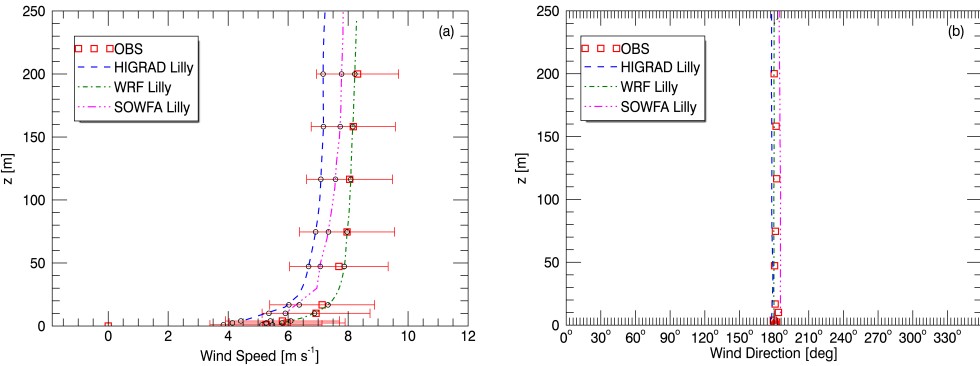

**Figure 14: Measured and simulated wind speed (a) and wind direction (b), using all three solvers, each with its optimal aspect ratio, during the convective case study.**

Figure 15 plots variability of the ten-minute average wind speed values from each simulation, with the measurement variability bars signifying one standard deviation from the mean value over all two hours of the simulation. All three models capture a similar range of wind speed variability as was observed, in contrast to the neutral case described above, with good agreement for the convective case attributed to the models' ability to accurately capture convective turbulent structures

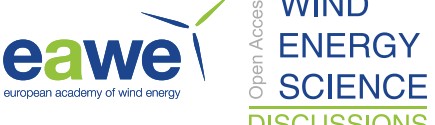



including updrafts and downdrafts, relatively steady geostrophic wind and surface flux forcing, and an absence of tower

wake contamination given the more southerly mean wind direction.

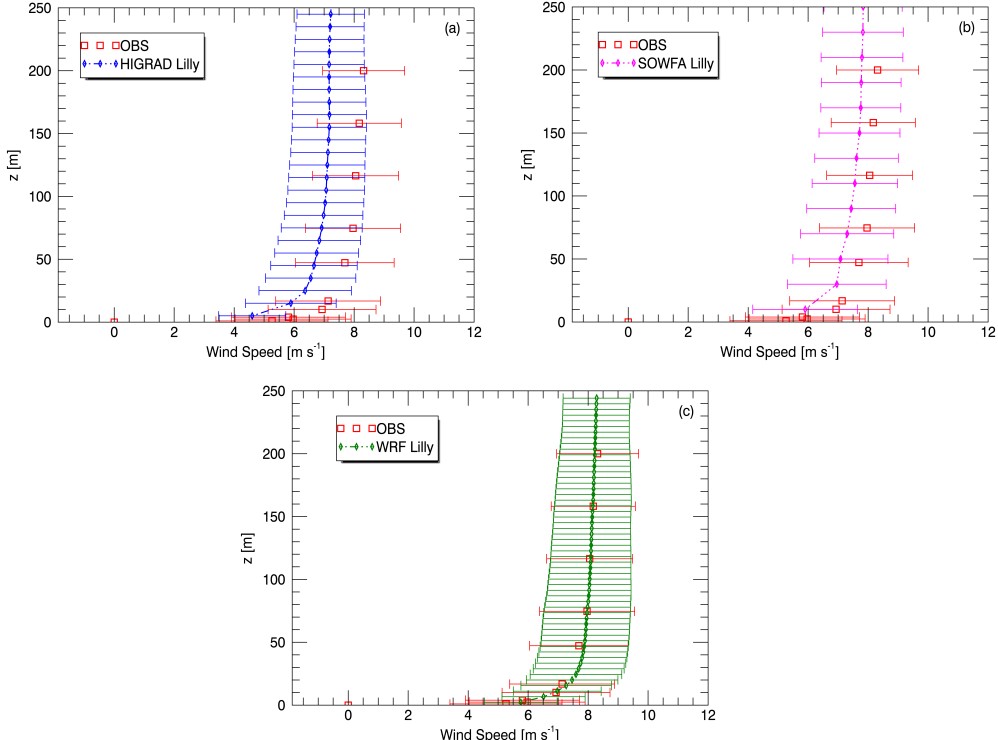

**Figure 15: Comparison of variability in observed and simulated wind speed profiles using all three solvers, HiGrad (a), SOWFA (b) and WRF (c), each with its optimal aspect ratio, during the convective case study.**

Quantitative MAE and RMSE scores, as in previous tables, from the convective simulations, are provided in Table 7.

Computed values of MAE and RMSE over all tower levels or over the turbine rotor disk confirm our previous observation of

excellent agreement between the WRF simulation results and measurements. Due to a well mixed layer characteristic of

convective ABLs, the shear over the rotor of a wind turbine is nearly zero and this is captured well by the models.

Assessment of turbulent quantities including TKE, stresses, and here sensible heat flux, was again carried out as

described for the neutral conditions, here over two hour period. Figure 16 shows measured versus simulated (resolved +

SFS) TKE (top left), streamwise vertical stress (top right) and vertical sensible heat flux (bottom) using each solver.



SOWFA provides the closest agreement of predicted TKE with the observations, with HiGrad and WRF predicting somewhat smaller values at all heights, but especially near the surface. Turbulent stress and sensible heat fluxes both show significant variability with height, with all model results agreeing broadly with the measurements. While the observed sensible heat fluxes are based upon virtual potential temperature, $\theta_v = \theta(1 + 0.61q_v)$ with $q_v$ the water vapor mixing ratio,

the simulations were dry. Hence, while exact comparison of simulated and observed heat fluxes is not possible, the dry conditions during the case study minimize discrepancies between these quantities.

**Table 7. Analysis of high-resolution LES performance using different models with the Lilly SFS parameterization for the convective ABL observed on August 17, 2012.**

|         | Tower MAE | Tower RMSE | Rotor MAE | Rotor Shear MAE |
|---------|-----------|------------|-----------|-----------------|
| HiGrad  | 1.24840   | 1.27849    | 1.54526   | 0.0339170       |
| WRF     | 0.138578  | 0.168826   | 0.219473  | 0.0637684       |
| SOWFA   | 0.570578  | 0.613874   | 0.411401  | 0.269006        |

As in the neutral case, spectra and cospectra are again computed, here by dividing two-hour time series into overlapping twenty-minute intervals (overlapping over ten minutes) and averaging the resulting eleven spectra. Figure 17 shows observed and simulated spectra of the streamwise velocity (top left), vertical velocity (top right), as well as observed $\theta_v$ and simulated $\theta$ spectra (bottom). For the velocity, agreement between the simulated and observed lower-frequency spectral content is good, with the expected attenuation of higher-frequency content from the simulated spectra due to filtering effects of the grid

and numerics, as described above. Spectra of temperature variables show greater low-frequency power from the simulations, possibly due to the specified value of $H_S$ being greater than the actual values (which are not available).

Figure 18 shows cospectra of the vertical velocity with the streamwise (upper left) and spanwise (upper right) components, as for the neutral case, along with those of the measured vertical velocity and $\theta_v$, versus simulated vertical velocity and $\theta$ (bottom). As with the spectra, measured and simulated cospectra likewise show good agreement, including

the sensible heat flux cospectrum, even though simulated temperature variance was higher than that of the observed $\theta_v$ (see Fig. 17c).





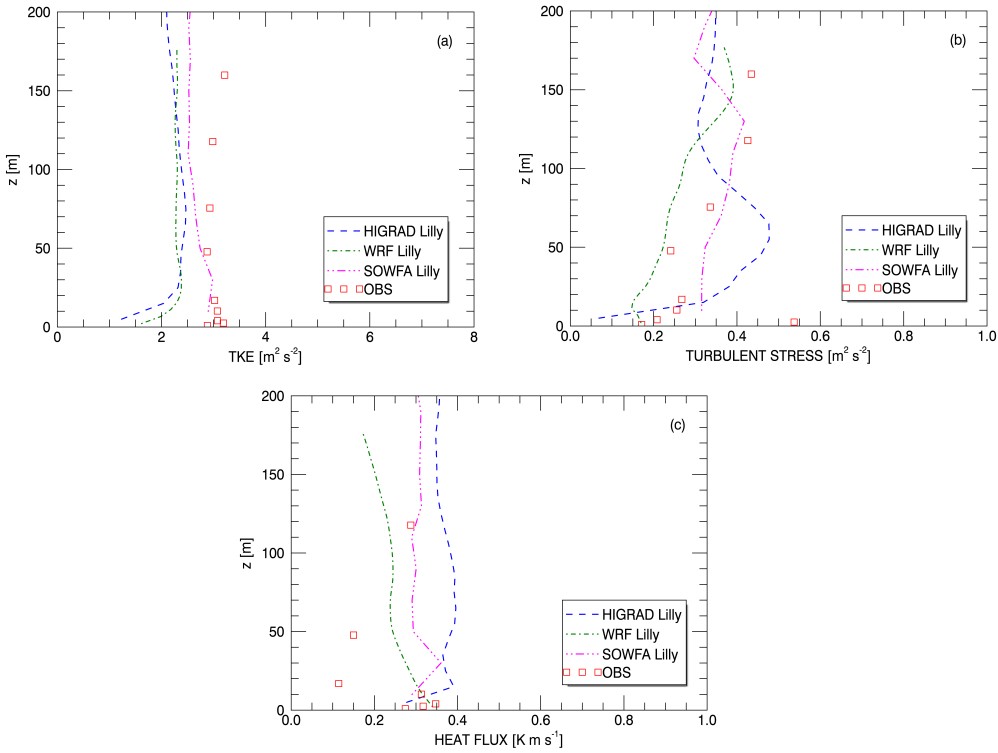

**Figure 16: Simulated and measured TKE (a), turbulent stress (b) and sensible heat flux (c), from the convective case study.**

**4 Summary and conclusions**

With a view toward assessing the applicability of idealized LES to provide turbulent flow quantities of interest to wind

5    power applications, three different LES solvers were utilized to simulate quasi-steady neutral and convective ABL flow

regimes. Simulations were compared against observations over nearly flat and homogeneous surface cover, for two case

studies featuring nearly-steady near-neutral and convective conditions, permitting use of idealized geostrophic forcing,

surface conditions, and periodic LBCs. The three solvers, encompassing a range of common numerical formulations and

turbulence SFS models, were subject to a series of sensitivity experiments to assess the impacts of variations of model

10   configuration and forcing parameters on quantities of interest, including wind speed, turbulent stresses and fluxes, TKE,

spectra and cospectra.

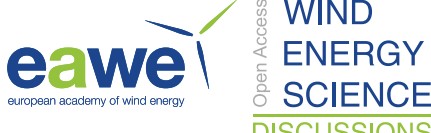



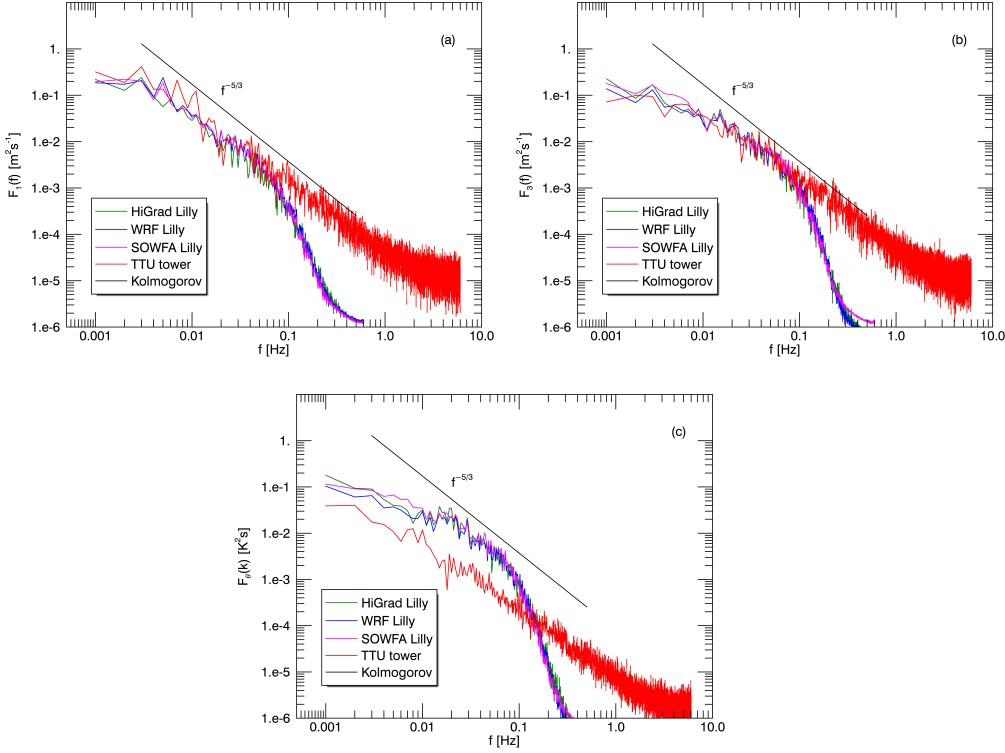

**Figure 17: Simulated and measured spectra of streamwise velocity (a), vertical velocity (b) and measured virtual potential temperature versus simulated vertical potential temperature (c), from the convective case study.**

5  A unique aspect of this study was computation of model turbulence statistics, spectra and cospectra, in the frequency

domain, enabling direct comparison with observed values. Spectral characteristics from all simulations displayed expected

qualitative characteristics, including peak energy at low wavenumbers, an inertial cascade, and attenuation of power with

increasing frequency. The narrower inertial subranges exhibited by the simulated versus the observed flows were due in part

to lower sampling rates of the simulations, and in part to the implicit model filter imposed by the mesh and numerical

10  discretization, the latter evidenced by the slightly wider inertial subranges from the SOWFA simulations.





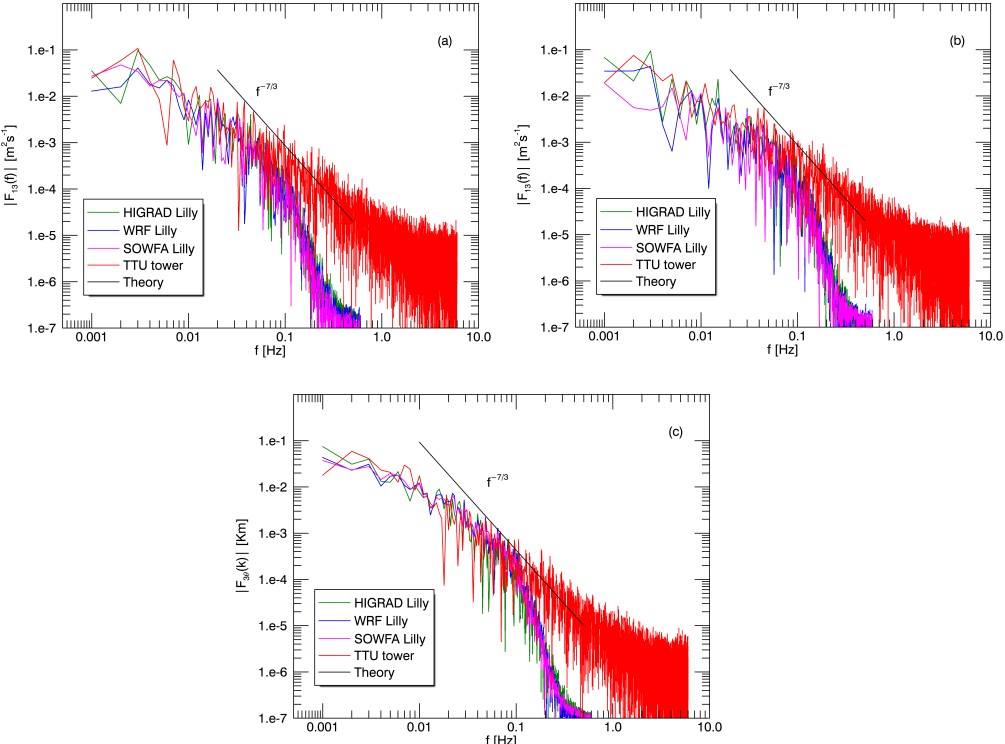

**Figure 18: Simulated and measured cospectra of vertical and streamwise velocity (a), vertical and spanwise velocity (b) and measured vertical velocity and virtual potential temperature versus simulated vertical velocity and potential temperature (c), from the convective case study.**

Comparison with observations reveals generally good performance of all models, under a typical range of configuration and forcing variations, supporting the use of idealized LES to produce useful flow and turbulence parameters during appropriate quasi-canonical flow conditions. The convective simulations provided generally better agreement with the observations, especially in quantities expressing variability, with superior performance attributed primarily to buoyancy-

10  generated turbulence dominating other forcing. While it is difficult to attribute the sources of discrepancies to features of the forcing versus generic limitations of the models, given the limitations of the data and simplicity of the model setups and forcing, sensitivity to different advection schemes, SFS parameterizations, and forcing was evident. An important conclusion is that the choice of advection discretization can be as important as the SFS parameterization.



Given the generally good performance of the idealized LES evaluated herein for simulating canonical, quasi-ideal cases, future efforts will focus on identifying sources of the discrepancies between the simulations and the observations noted herein. One approach will be to conduct mesoscale simulations of quasi-ideal case studies such as examined here to obtain estimates of various forcing parameters not available from the observations, or representable using constant values, such as

5 changes of $U_g$ over time, and advections of momentum and temperature. Incorporation of these additional forcing parameters may enable quasi-idealized simulations to capture a wider range of meteorological conditions. Full coupling of microscale and mesoscale simulations will also be pursued, with a view toward creation of a full-spectrum simulation capability applicable to arbitrary conditions. The present study provides a necessary first step and background support for future assessment of more general mesoscale to microscale coupling techniques.

**Acknowledgements.** This work was performed under the auspices of the U.S. Department of Energy (DOE) by Lawrence Livermore National Laboratory, the National Renewable Energy Laboratory, Los Alamos National Laboratory, Pacific Northwest National, Argonne National Laboratory, and Sandia National Laboratories, under contracts DE-AC52-07NA27344, DE-AC36-08GO28308, DE-AC52-06NA25396, DE-A06-76RLO 1830, DE-AC02-06CH11357 and DE-

15 AC04-94AL85000, respectively. Ramesh Balakrishnan used resources of the Argonne Leadership Computing Facility, which is a DOE Office of Science User Facility supported under Contract DE-AC02-06CH11357. All participants were supported by the US DOE Office of Energy Efficiency and Renewable Energy.




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
