# Peer review of "Large-Eddy Simulation Sensitivities to Variations of Configuration and Forcing Parameters in Canonical Boundary-Layer Flows for Wind Energy Applications"

_Wind Energy Science, 2017_

## Referee Comment (RC1) · Anonymous Referee #1 · 8 Nov 2017

**General comments:**

The paper analyses the sensitivity of large-eddy simulations to various model configuration and physical forcing parameters. Simulations results of three different LES models are compared among each other as well as to mast measurements. The objective is to prove that LES are a valid tool to provide flow parameters relevant to wind energy applications.

Although LES as a tool for assessing wind energy relevant flow parameters is not novel

by itself – at least in the scientific community – the intercomparison of several LES models and typical model configurations is a valuable contribution to the community. Even more so as the simulation results are being validated with observations the lack of which is very often the weak point of such studies.

The paper is in general well organized and clearly written. I can recommend it for publication after some minor revisions.

**Specific comments:**

Section 1: Aren't there any similar studies (comparison of different LES models and/or model configurations for atmospheric flows) in the available literature? If there aren't, state that to strengthen your paper. If there are any somehow similar studies, mention them.

Section 2.1: Why did you not investigate a stable case? Is there any particular reason for it? Because evidently, stable regimes are very important for wind energy.

Section 2.1: Elaborate a bit more on the measurement data used to validate the simulation results. It reads as if the SWiFT tower is only 50 m high, but I guess it is 200 m high (if not: where does the data above 50 m come from?).

Section 2.1: How did you estimate the geostrophic wind speed from the wind profiler measurements? Which additional uncertainty is introduced by this method?

Page 9, lines 19 ff.: The simulation results are evaluated when the average flow velocity has reached its first maximum. Hence, the simulation is not steady at that point, it will take much longer time until the inertial oscillation has been damped completely. Nevertheless, your approach is valid as such long simulations are not feasible and the important point is that all the simulations are compared at the same state of evolution. You could just clarify this a bit more.

Page 10, line 9: How did you estimate the "representative" values of HS? From literature, previous experience or measurements?

Section 2.6, Tables 1 and 2: Please be clearer on which parameters come directly from measurements and which are just estimated or set arbitrarily.

Tables 1 and 2: Units are missing!

Figure 3: It appears as if the boundary layer height is considerably different in the three simulations (ca. 1600-1800 m in WRF, 1400-1600 m in SOWFA, 1200-1400 m in HiGrad). Can you comment on that?

Page 20, line 7: Something is missing here, I think you mean RMSE and MAE at the tower (see Tables 3, 4). And furthermore: What does "Tower MAE" or "Tower RMSE" mean? At the (imaginary) turbine tower or at the met mast? In specific heights or averaged over all heights? Looking at page 26, I find the answer, but that should be explained here, too and in more detail.

Page 20, last sentence: That is not exactly correct, as Table 3 shows that the rotor MAE is considerably higher for the higher-order advection scheme.

Page 22, line 13: "The WRF model shows the greatest sensitivity to these parameters with HiGrad showing the least." I don't agree: From Fig. 9 it looks as if WRF (bottom left) shows the least variability whereas SOWFA shows the greatest (middle left).

Fig. 10: Add in the caption that these are the runs with higher resolution. Otherwise, this figure cannot be distinguished from previous figures.

Page 24: The subsection is called "Sensitivity to model grid resolution" but actually only the higher resolution runs are compared among each other. The comparison to the lower resolution runs is missing.

Page 26, lines 9 ff.: The TKE discrepancy is quite huge, even if compared with the unwaked case. Can you cite any studies from the literature that provide estimates of

tower wake effects on TKE? To my experience the effect should be large but not factor 3.

Section 3.2.2: WRF is reproducing the measurements amazingly perfect including the variability. Have you investigated a second convective case? Otherwise you cannot be sure if this is just a fluke or if WRF is generally that well suited for convective conditions. Can you comment on this?

Page 32, line 16: What was the result of the sensitivity test towards the surface heat flux? Does it confirm the guess that $H_S$ was possibly too large in the simulations?

The summary/conclusions section is quite short. Which settings turned out to be the best (or better than others)? Can you give any recommendations?

Page 36, lines 1-5: These are indeed the next necessary steps. In these potential follow-up studies you should – in my view – strongly separate physical forcing and numerical parameters. Even in this manuscript they are from my point of view too much intertwined. The paper could benefit from some more separation between forcing and numerical parameters.

**Technical corrections:**

Page 4, line 5: "focuses" –> "focus"

Section 2: All the subsections need to be renumbered (1.1 –> 2.1 etc.)

Page 9, line18: "Sect. 1.3" –> "Sect. 2.3"

Page 13, line 1: "plan" –> "planar" or "plane" and "x-z" –> "x-y"

Page 13, line 7: "used grid resolution" –> "used a grid resolution"

Page 17, line 16: "Sect. 1.6" –> "Sect. 2.6"

Page 22, line 9: $U_g$ is not 9 m/s but 6.5 m/s in the neutral case (according to Table 1).

Page 24, line 7: $U_g$ is not 9 m/s but 6.5 m/s in the neutral case (according to Table 1).

Page 26, line 22: "August 04 case" –> "August 17 case"

Fig. 16: Switch the two upper plots to make it consistent to Fig. 11 (or switch there).

Figures in general: Try to improve the readability of the figures by increasing font size, make lines slightly thicker, show only relevant parts etc. The dash patterns are sometimes very similar so that the curves can hardly be distinguished on a b/w print.

Fig. 11: Introduce different dash patterns to allow distinguishing the curves on a b/w print.

Page 31, line 12: insert "a" between "over" and "two"

---

## Referee Comment (RC2) · Anonymous Referee #2 · 7 Dec 2017

Dear Authors,

The article is very interesting and is overall well formulated. It does approach a very interesting and important research area. It further contributes with important results, comparing a number of models, interesting for the research community.

The performed simulations are performed for specific stability cases, please explain why these cases have been selected and others have been left out and to what extent these cases are relevant for the field. Perhaps the cases are choses do to available

measurements data?

There are parts of the text where clarifications could improve the quality and readability of the article. For example, further explanation regarding the used measurement data and also the comparison between the measurements and LES data, is needed.

I recommend this article for publication after minor revision based on the above comments.

Best regards

---

## Author Comment (AC1) · 12 Jan 2018

General comments:

The paper analyses the sensitivity of large-eddy simulations to various model configuration and physical forcing parameters. Simulations results of three different LES models are compared among each other as well as to mast measurements. The objective is to prove that LES are a valid tool to provide flow parameters relevant to wind energy applications.

Although LES as a tool for assessing wind energy relevant flow parameters is not novel by itself – at least in the scientific community – the intercomparison of several LES models and typical model configurations is a valuable contribution to the community. Even more so as the simulation results are being validated with observations the lack of which is very often the weak point of such studies. The paper is in general well organized and clearly written. I can recommend it for publication after some minor revisions.

*The authors wish to thank the reviewer for a careful proofreading that uncovered several errors that would have ranged from annoying to confusing, and also identifying areas that would benefit from clarification. The manuscript is much improved due to these valuable suggestions. Reviewer comments appear in* Microsoft New Tai Lue*, our responses are in Calibri Italic, text from the paper is in* "times new roman"*, enclosed by quotes, with new additions to the existing text in* **bold American typewriter***.*

Specific comments:

1.      Section 1: Aren't there any similar studies (comparison of different LES models and/or model configurations for atmospheric flows) in the available literature? If there aren't, state that to strengthen your paper. If there are any somehow similar studies, mention them.

*The uniqueness of the present study and where it fits in relation to other published work (of which we are aware) has been incorporated into the last paragraph of Section 1 (beginning page 3, line 21; with several new references), which now reads:*

"The present study, conducted under the auspices of the A2e MMC project, examines the efficacy of idealized atmospheric LES using periodic lateral boundary conditions (LBCs)**, an approach commonly applied in fundamental and applied ABL studies (see e.g. Deardorff, 1970; Deardorff, 1980; Moeng, 1984; Kosović and Curry, 2000),** to provide flow parameters of interest to wind energy applications. **The present study is unique in its focus on the representation of the accuracy of the simulated flow, rather than on turbine interactions, including detailed comparison of simulated and observed turbulence information, including spectra and cospectra. Moreover, we investigate this capability using a computational framework that is both relatively mature and reasonably economical, in comparison with more general yet also more complicated and expensive methods, such as those incorporating time varying mesoscale input via additional internal forcing terms (e.g. Sanz Rodrigo et al. 2017) or at the lateral domain boundaries (e.g. Muñoz-Esparza et al., 2017; Rai et al. 2017ab). Finally, an** examination of uncertainties **provides** a required basis for assessment of both existing idealized

simulation capabilities, as well as of more sophisticated MMC techniques under development**, to the wind energy arena**.

2.      Section 2.1: Why did you not investigate a stable case? Is there any particular reason for it? Because evidently, stable regimes are very important for wind energy.

*We agree that stable conditions are of high importance to wind energy. We did investigate a stable case study, but found it difficult to reproduce important low-level jet characteristics, given the strong susceptibility of stable flows to subtle forcings, such as baroclinicity and advections, which were not considered within the study. In addition, the extreme computational demands of simulating stable conditions, given both the high resolution required to capture turbulence, and the long durations to capture the decay of the afternoon ABL and the inertial oscillation, were such that we felt that a study of stable conditions deserved its own paper, where these issues could be more fully explored. Therefore, a similar study devoted to simulating the stable ABL is reserved for future work. To address the absence of a stable case study in the present study, section 2.1 was rewritten (text beginning on P5 Line 10):*

"To satisfy conditions under which idealized forcing is appropriate, data were examined for case studies encompassing canonical **ABL regimes occurring during convective, neutral and stable conditions**. Criteria for case selection included **nearly constant values of wind speed, wind direction, and $H_S$, over time windows of a few hours,** with minimal mesoscale variability and **influences of moist processes.**

Several periods approximating quasi-canonical convective ABL conditions **were found** within the observational data**, with the most ideal, that occurring during the apex of solar heating during the early afternoon of July 4, 2012, selected for the convective case study. In contrast, canonical neutral conditions occurred relatively infrequently, and for much shorter durations, during evening and morning transitions. As transitional boundary layers contain the imprint of preceding stable or convective forcing, many of the candidate neutral periods showed strong influence from prior states and thus were not considered.** Furthermore, sonic anemometers on the meteorological tower are mounted on the booms pointing in the West-northwest direction while the dominant wind direction at the SWiFT tower is Southerly. As such, most of the candidate neutral cases occurred during times at which the instruments were influenced somewhat by the tower wake. **With respect to these constraints, the optimal neutral case study** occurred during the evening transition of August 17, 2012.

**While stable conditions are of high importance for wind energy, the combination of difficulties in specifying proper forcing (to capture the correct evolution of the nocturnal low-level jet) and the high computational demands imposed by the fine mesh resolutions required to capture sustained turbulence during moderately stable nocturnal conditions precluded inclusion of a stable case study in the present study.**"

3.      Section 2.1: Elaborate a bit more on the measurement data used to validate the simulation results. It reads as if the SWiFT tower is only 50 m high, but I guess it is 200 m high (if not: where does the data above 50 m come from?).

*We apologize for this error. There are several towers in and around the SWiFT site, however the data used in this study were obtained from only the 200 m tower, and a RASS. We have modified the text to clarify as follows (text beginning on P4 Line 19):*

"The Sandia Scaled Wind Farm Technology (SWiFT) test facility, located in the US Southern Great Plains, was selected for the study, due to its nearly flat terrain and homogeneous surface cover, permitting reasonable approximation in idealized computational setups consisting of flat terrain with uniform surface characteristics and forcing conditions, as well as periodic LBCs. **Data used to force and evaluate the simulations were obtained from two instrument platforms, a 200 m instrumented meteorological tower, and a radio acoustic sounding system (RASS), each located at the neighboring Texas Tech University's National Wind Institute (NWI). The tower provided fast-response data at ten heights between 0.9 and 200 m from which turbulence and mean flow data were computed, while the RASS data provided assessment of** the prevailing meteorology, as well as estimates of a common parameter used to force atmospheric LES, the geostrophic wind speed, $U_g = \sqrt{u_g^2 + v_g^2}$, with $u_g$ and $v_g$ denoting zonal and meridional components, respectively. **Values of $u_g$ and $v_g$ were estimated using RASS data from above the ABL top, then adjusted slightly until the simulated wind speed and direction profiles above the ABL closely matched the observed values**. Other parameters required to force the simulations include the roughness length, $z_0$, which was estimated from the land cover, and fluxes of sensible heat, $H_S$, estimated **using values computed from the lowest sensors on the tower, beginning at 0.9 m, and bulk Richardson numbers computed between the 2.4 and 10.1 m measurement heights (see Kelly and Ennis (2016) for further information about the instrumentation and site characteristics)."**

4.      Section 2.1: How did you estimate the geostrophic wind speed from the wind profiler measurements? Which additional uncertainty is introduced by this method? Page 9, lines 19 ff.: The simulation results are evaluated when the average flow velocity has reached its first maximum. Hence, the simulation is not steady at that point, it will take much longer time until the inertial oscillation has been damped completely. Nevertheless, your approach is valid as such long simulations are not feasible and the important point is that all the simulations are compared at the same state of evolution. You could just clarify this a bit more.

*The method of estimation of the geostrophic wind speed was clarified in Sect. 2.1, as described in the response to the previous comment. In addition, the following text was added (beginning on Page 10, line 14):*

"**While the flow had not equilibrated completely, this methodology allowed for comparison at the same point in the evolution of each simulation. Further, wind speed values at 80 m varied by only a few tenths of a m s$^{-1}$ over the period spanning the peak, yielding minimal impacts on quantities of interest. Continuing the simulations further in time would have achieved only negligible changes at the expense of reducing the number of configurations examined, given the high computational expense of each simulation.**"

5.      Page 10, line 9: How did you estimate the "representative" values of HS? From literature, previous experience or measurements?

*A fuller description of the parameter estimation values is now provided in Sect. 2.1 (see response to comment 3). The text at this location now reads (beginning on page 11, line 2):*

"Sensitivity to forcing was examined by varying $U_g$ and $z_0$ around **estimated base values (as described in Sect. 2.1) during the neutral case study, and using two representative values of $H_S$, with $U_g$ and $z_0$ values held constant, during convective conditions**."

6.        Section 2.6, Tables 1 and 2: Please be clearer on which parameters come directly from measurements and which are just estimated or set arbitrarily.

*Only the atmospheric and surface parameters within the tables are estimated. Therefore, rather than modifying the tables, the missing information is provided in the text describing the tables, as elucidated in the response comment 3.*

7.        Tables 1 and 2: Units are missing!

*Units added.*

8.        Figure 3: It appears as if the boundary layer height is considerably different in the three simulations (ca. 1600-1800 m in WRF, 1400-1600 m in SOWFA, 1200-1400 m in HiGrad). Can you comment on that?

*The following statement was added (beginning page 16, line 18):*

"**WRF also generates a slightly deeper ABL, likely due to a combination of higher vertical resolution and a warmer ABL, the latter slightly reducing the relative strength of the capping inversion. HiGrad results in the shallowest ABL, despite using the same resolution as SOWFA, likely due to its use of an odd-order advection operator, which being more dispersive than SOWFA's even order operator, slightly reduces TKE (see also Fig. 16a).**"

9.        Page 20, line 7: Something is missing here, I think you mean RMSE and MAE at the tower (see Tables 3, 4). And furthermore: What does "Tower MAE" or "Tower RMSE" mean? At the (imaginary) turbine tower or at the met mast? In specific heights or averaged over all heights? Looking at page 26, I find the answer, but that should be explained here, too and in more detail.

*This has been clarified and now reads (page 21 line 5):*

"**The relative performances of various configurations are assessed quantitatively using the Mean Absolute Error (MAE), Root Mean Square Error (RMSE) and vertical shear, computed across two different depths. Tower MAE and RMSE were computed over all heights on the tower spanned by the model mesh (no extrapolation to tower values below the lowest model height), by interpolating model values to the sensor heights using cubic splines. Rotor MAE and shear were computed analogously over a depth of 40 to 140 m, corresponding to the swept are of a representative modern utility-scale wind turbine with a 100 m rotor diameter and a hub height of 90 m. Wind profile characteristics within and across the rotor swept area are relevant to both power production and fatigue loading.**"

*In addition, the redundant description previously occurring on page 26 was removed.*

10.     Page 20, last sentence: That is not exactly correct, as Table 3 shows that the rotor MAE is considerably higher for the higher-order advection scheme.

*We agree. This sentence has been changed to (page 22 line 1):*

"Table 3 shows **the impact of varying the order of the advective operators within the HiGrad model on each of the above statistics, indicating that changes to this configuration choice result in generally small changes in velocity profile characteristics across both the tower and the rotor, with neither the higher- nor lower-order results notably superior overall**."

11.     Page 22, line 13: "The WRF model shows the greatest sensitivity to these parameters with HiGrad showing the least." I don't agree: From Fig. 9 it looks as if WRF (bottom left) shows the least variability whereas SOWFA shows the greatest (middle left).

*Your observation is correct; the sentence was removed.*

12.     Fig. 10: Add in the caption that these are the runs with higher resolution. Otherwise, this figure cannot be distinguished from previous figures.

*Done.*

13.     Page 24: The subsection is called "Sensitivity to model grid resolution" but actually only the higher resolution runs are compared among each other. The comparison to the lower resolution runs is missing.

*The sub-section name was changed to "***Assessment of high-resolution simulations***"*

14.     Page 26, lines 9 ff.: The TKE discrepancy is quite huge, even if compared with the unwaked case. Can you cite any studies from the literature that provide estimates of tower wake effects on TKE? To my experience the effect should be large but not factor 3

*We are not aware of any published literature on the magnitude of tower wake effects on TKE. We agree that the effects are quite large. Figures 2.2 and 2.4 in the SWiFT site characterization report (Kelley and Ennis, 2016, which is cited in the manuscript) show the effect of the tower wake on TI, with increase 2.5-3 times which is particularly clear in the neutral and stable ABL states.*

15.     Section 3.2.2: WRF is reproducing the measurements amazingly perfect including the variability. Have you investigated a second convective case? Otherwise you cannot be sure if this is just a fluke or if WRF is generally that well suited for convective conditions. Can you comment on this?

*We did not investigate a second convective case. WRF agrees slightly more closely with the data than the other models for this simulation, however all solvers provide mean values that are well within the variability, and ranges of values that also agree well. Given that several forcing parameters were estimated, we do not conclude that WRF is better than the other solvers, only that they all do very well in this convective case study.*

16.      Page 32, line 16: What was the result of the sensitivity test towards the surface heat flux? Does it confirm the guess that HS was possibly too large in the simulations?

*We were remiss in not indicating that we presented results from the simulations only using the lower of the two surface heat flux values, as those provided superior agreement with the observations, while already over-predicting the potential temperature spectra (Fig 17c). We added the following sentence at the beginning of Sect. 3.2.2 which introduces the convective case study (page 32 line 7):*

   "**All results presented herein are from simulations using the smaller of the two values of** $H_S = 0.35$ **K m s$^{-1}$, corresponding to approximately 400 W m$^{-2}$.**"

*Given the already large number of sensitivities examined in the paper, we did not address the impacts of changing HS.*

17.      The summary/conclusions section is quite short. Which settings turned out to be the best (or better than others)? Can you give any recommendations?

*The major point we wish to emphasize is that for relatively steady forcing and flow conditions, these idealized LES all provide valuable information for various wind energy applications, irrespective of relatively minor changes to the numerics, computational domain setup, or physical forcing parameters. This is reflected in the sentence,* "Comparison with observations reveals generally good performance of all models, under a typical range of configuration and forcing variations, supporting the use of idealized LES to produce useful flow and turbulence parameters during appropriate quasi-canonical flow conditions." *(Page 36, Line 6). Beyond that, we do not feel confident in giving any recommendations, given uncertainties in the estimates of key forcing parameters.*

18.      Page 36, lines 1-5: These are indeed the next necessary steps. In these potential follow-up studies you should – in my view – strongly separate physical forcing and numerical parameters. Even in this manuscript they are from my point of view too much intertwined. The paper could benefit from some more separation between forcing and numerical parameters.

*We are sympathetic to the criticism that better separation of different factors would have been useful. However, given the expense of conducting the large number of LES comprising the study, and our goal of a concise and comprehensible presentation of results, some consolidation was necessary. Moreover, this study provides a foundation for follow-up work to further separate the impacts of various factors using similar setups that we now know work.*

*We have added a few phrases to the final paragraph to elucidate this point (beginning page 38, line 1):*

"Given the generally good performance of the idealized LES evaluated herein for simulating canonical, quasi-ideal cases, future efforts will focus on further identifying sources of the discrepancies between the simulations and the observations, **including further isolation of the impacts of choices of numerical methods, domain configuration, and physical forcing parameter values on various quantities of interest**. One approach will be to conduct mesoscale simulations of quasi-ideal case studies such as examined here to obtain better estimates of various forcing parameters not available from the observations, or representable using constant values, such as changes of $U_g$ over time, and advections of momentum and temperature. Incorporation of these additional forcing parameters may enable quasi-idealized simulations to capture a wider range of meteorological conditions**, and also**

**enable further elucidation of the roles of numerical and configuration changes in simulation accuracies**. Full coupling of microscale and mesoscale simulations will also be pursued, with a view toward creation of a full-spectrum simulation capability applicable to arbitrary conditions. The present study provides a necessary first step and background support for future assessment of more general **and robust** mesoscale to microscale coupling techniques."

Technical corrections:

1.      Page 4, line 5: "focuses" –> "focus"

   *Done*

2.      Section 2: All the subsections need to be renumbered (1.1 –> 2.1 etc.)

   *Done. Also, further subsections within Sect. 3 were added for improved delineation.*

3.      Page 9, line18: "Sect. 1.3" –> "Sect. 2.3"

   *Done.*

4.      Page 13, line 1: "plan" –> "planar" or "plane" and "x-z" –> "x-y"

   *Done.*

5.      Page 13, line 7: "used grid resolution" –> "used a grid resolution"

   *Done.*

6.      Page 17, line 16: "Sect. 1.6" –> "Sect. 2.6"

   *Done.*

7.      Page 22, line 9: Ug is not 9 m/s but 6.5 m/s in the neutral case (according to Table 1).

   *Done.*

8.      Page 24, line 7: Ug is not 9 m/s but 6.5 m/s in the neutral case (according to Table 1).

   *Done.*

9.      Page 26, line 22: "August 04 case" –> "August 17 case"

   *Done.*

10.     Fig. 16: Switch the two upper plots to make it consistent to Fig. 11 (or switch there).

   *Switched Stress and TKE in Fig. 16 to match Fig. 11.*

11.	Figures in general: Try to improve the readability of the figures by increasing font size, make lines slightly thicker, show only relevant parts etc. The dash patterns are sometimes very similar so that the curves can hardly be distinguished on a b/w print.

*New figures have been uploaded to improve clarity and distinguishability.*

12.	Fig. 11: Introduce different dash patterns to allow distinguishing the curves on a b/w print.

*Done.*

13.	Page 31, line 12: insert "a" between "over" and "two"

*Done.*

*Several other small changes were made throughout the text to improve clarity and readability.*

---

## Author Comment (AC2) · 12 Jan 2018

The article is very interesting and is overall well formulated. It does approach a very interesting and important research area. It further contributes with important results, comparing a number of models, interesting for the research community. The performed simulations are performed for specific stability cases, please explain why these cases have been selected and others have been left out and to what extent these cases are relevant for the field. Perhaps the cases are choses do to available measurements data?

There are parts of the text where clarifications could improve the quality and readability of the article. For example, further explanation regarding the used measurement data and also the comparison between the measurements and LES data, is needed. I recommend this article for publication after minor revision based on the above comments.

*The authors thank the reviewer for the favorable assessment. We hope that the clarifications presented in response to reviewer 1's specific comments (particularly comment 2) are sufficient.*

*Several other small changes were made throughout the text to improve clarity and readability.*